# Bicarbonate signalling via G protein-coupled receptor regulates ischaemia-reperfusion injury

Airi Jo-Watanabe [1,2] ✉, Toshiki Inaba[3], Takahiro Osada [4], Ryota Hashimoto [5], Tomohiro Nishizawa [6], Toshiaki Okuno[1], Sayoko Ihara[7], Kazushige Touhara[7], Nobutaka Hattori [3,8], Masatsugu Oh-Hora [1,5], Osamu Nureki [9] & Takehiko Yokomizo [1] ✉

Homoeostatic regulation of the acid–base balance is essential for cellular functional integrity. However, little is known about the molecular mechanism through which the acid–base balance regulates cellular responses. Here, we report that bicarbonate ions activate a G protein-coupled receptor (GPCR), i.e., GPR30, which leads to $G_q$-coupled calcium responses. *Gpr30-Venus* knock-in mice reveal predominant expression of GPR30 in brain mural cells. Primary culture and fresh isolation of brain mural cells demonstrate bicarbonate-induced, GPR30-dependent calcium responses. GPR30-deficient male mice are protected against ischemia-reperfusion injury by a rapid blood flow recovery. Collectively, we identify a bicarbonate-sensing GPCR in brain mural cells that regulates blood flow and ischemia–reperfusion injury. Our results provide a perspective on the modulation of GPR30 signalling in the development of innovative therapies for ischaemic stroke. Moreover, our findings provide perspectives on acid/base sensing GPCRs, concomitantly modulating cellular responses depending on fluctuating ion concentrations under the acid–base homoeostasis.

The acid–base balance, a vital mechanism that maintains the optimal pH homoeostasis for cellular function, is based on multiple buffering systems, including the bicarbonate buffer system in vivo. Under normal physiological conditions, the pH in humans, on a systemic level, does not deviate from a narrow range (7.40 ± 0.05)[1]. In contrast, proton, bicarbonate ion, and carbon dioxide ($CO_2$) concentrations fluctuate locally in a variety of physiological and pathological conditions such as in gastric acid secretion[2], the tumour environment[3], and ischaemia-reperfusion injury in myocardial infarction[4] and stroke[5], in

which dynamic changes in local pH and ion homoeostasis regulate cellular responses[6].

Changes in the extracellular environment are monitored by membrane receptors, most of which are transient receptor potential channels and G protein-coupled receptors (GPCRs). Previously, we and other groups have established that several GPCRs are proton receptors[7–9]. These proton-sensing GPCRs respond to a reduction in extracellular pH (i.e. an increase in proton concentration), particularly in the acidic tumour microenvironment, at inflamed sites, and in

[1]Department of Biochemistry, Juntendo University Graduate School of Medicine, Tokyo 113-8421, Japan. [2]AMED-PRIME, Japan Agency for Medical Research and Development, Tokyo 100-0004, Japan. [3]Department of Neurology, Juntendo University School of Medicine, Tokyo 113-8421, Japan. [4]Department of Neurophysiology, Juntendo University School of Medicine, Tokyo 113-8421, Japan. [5]Laboratory of Cell Biology, Biomedical Research Core Facilities, Juntendo University Graduate School of Medicine, Tokyo 113-8421, Japan. [6]Graduate School of Medical Life Science, Yokohama City University, Kanagawa 230-0045, Japan. [7]Department of Applied Biological Chemistry, Graduate School of Agricultural and Life Sciences, The University of Tokyo, Tokyo 113-8657, Japan. [8]Neurodegenerative Disorders Collaborative Laboratory, RIKEN Center for Brain Science, Saitama 351-0198, Japan. [9]Department of Biological Sciences, Graduate School of Science, The University of Tokyo, Tokyo 113-0033, Japan. ✉e-mail: awatanabe-tky@umin.ac.jp; yokomizo-tky@umin.ac.jp

ischaemia-reperfusion injury[10]. Alternatively, recent studies have demonstrated that bicarbonate and $CO_2$ directly regulate various cellular responses independently of pH[11,12]. This finding led us to hypothesise that another acid–base balance-related GPCR modulates signal transduction in response to dynamic changes in the local acid–base balance.

In the present study, using public databases containing GPCR expression data in tissues and single cells, we first chose candidate GPCRs based on their specific expression in the stomach and pancreas, where acid or alkaline secretions immediately locally act on neighbouring cells under physiological conditions. Four of 10 GPCRs selected as the primary candidates were highly expressed in the brain. Pathological conditions that cause a dynamic shift in the acid–base balance include ischaemia and reperfusion, which directly affect vascular and perivascular cells via channels and receptors. Therefore, we subsequently searched for acid–base balance-related GPCRs expressed in the brain microvasculature. Finally, we investigated how the selected GPCR was involved in cerebral ischaemia-reperfusion injury, which primarily contributes to the pathophysiology of ischaemic stroke[13].

## Results

### GPR30 expression in local pH fluctuation

We comprehensively examined the expression of murine GPCRs in tissues using publicly available data from the Psychoactive Drug Screening Program (PDSP) database[14] (Supplementary Fig. 1a). Ten of 353 GPCRs examined were selected based on their predominant expression in the stomach and pancreas (Supplementary Fig. 1b). Tissue-specific expression was further confirmed using a second

database−BioGPS (http://biogps.org/). The candidate GPCRs included four receptors for gastrointestinal hormones and hormone-related peptides, three vasoactive receptors, one receptor that is both vasoactive and hormone-related, and two receptors that activate the Wnt signalling pathway. Four of the 10 candidate GPCRs were highly expressed in the brain (Supplementary Fig. 1c). We subsequently searched the expression of these 10 candidates in the brain microvasculature at the single-cell level. The single-cell RNA-sequencing database of mouse brain vascular and perivascular cells (http://betsholtzlab.org/VascularSingleCells/database.html)[15,16] distinguished *Gpr30* by its specific expression in the brain microvasculature (Supplementary Fig. 1d, e).

### Bicarbonate activates GPR30 in vitro

GPR30 is a G protein-coupled oestrogen receptor (GPER) that mediates the rapid non-genomic action of oestradiol ($E_2$)[17,18]. Despite accumulating evidence on the pleiotropic functions of GPR30 in vivo[19–21], there has been a controversy surrounding the response of GPR30 to $E_2$ and a putative synthetic agonist G-1 in vitro[22,23], ex vivo[24], and in vivo[25]. Furthermore, the tissue specificity of *Gpr30* expression (Supplementary Fig. 1b) does not preclude the possibility of non-oestrogenic functions of GPR30.

We first evaluated the non-genomic action of $E_2$ through GPR30. The calcium mobilisation (Fig. 1a) and TGFα shedding assays[26] (Supplementary Fig. 2a) did not demonstrate activation of GPR30 by $E_2$ in MCF-7 cells stably expressing human GPR30 (hGPR30, MCF-GPR30) and HEK293 cells transiently expressing hGPR30. $E_2$ caused a dotted accumulation of the nuclear oestrogen receptors (ERs) in COS-7 cells transfected with ERα-EGFP (Supplementary Fig. 2b), demonstrating

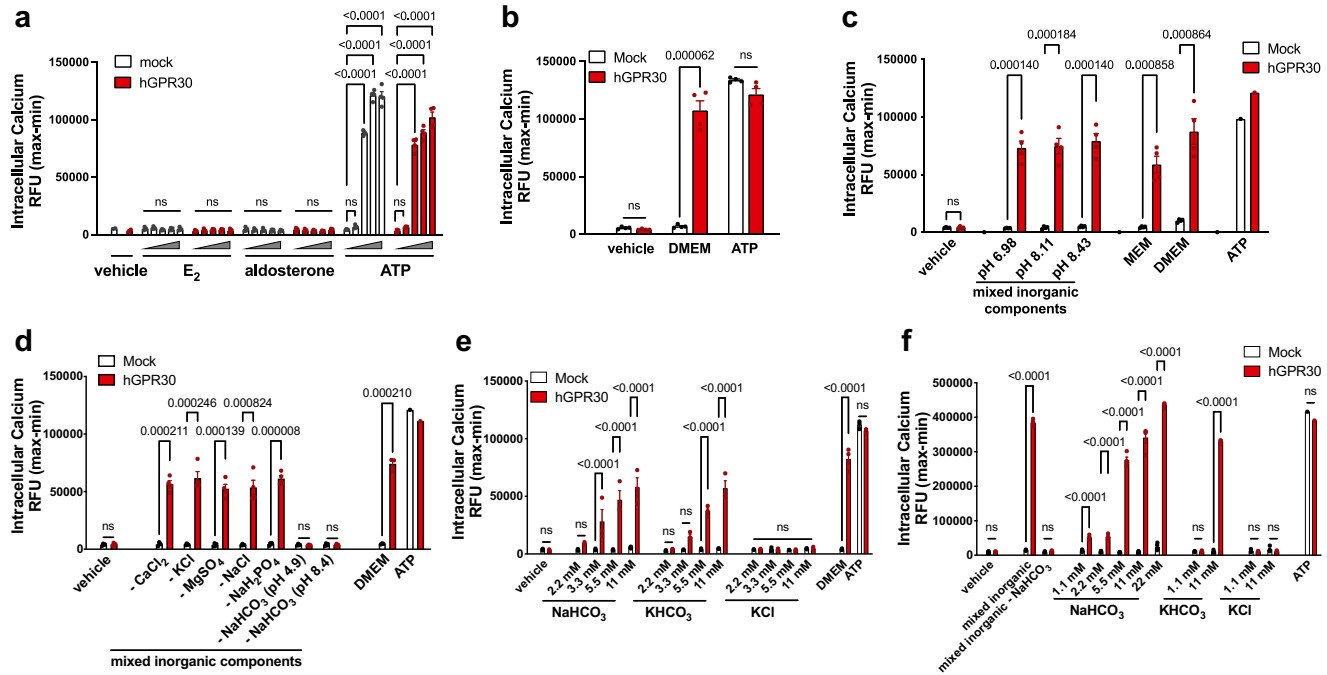

**Fig. 1 | Physiological concentration of bicarbonate ions activate GPR30 in vitro. a–f** Calcium mobilisation assay in Fluo-8-loaded MCF-7 (MCF-GPR30, a–e) and HEK293 (HEK-GPR30, f) cells stably expressing human GPR30 (hGPR30). The increase in the intracellular calcium level was evaluated by maximum relative fluorescence units (RFU) minus minimum RFU (max − min). ATP activates endogenous purinergic receptors and serves as a positive control. **a** Cells treated with vehicle, oestradiol ($E_2$, $10^{-11}$–$10^{-7}$ M), aldosterone ($10^{-11}$–$10^{-7}$ M), and ATP ($10^{-8}$–$10^{-4}$ M). **b** Cells treated with vehicle, Dulbecco's modified Eagle's medium (DMEM), and ATP (25 μM). **c** MCF-GPR30 cells were treated with the mixed inorganic solution at various pH values, Minimum Essential Medium (MEM), and DMEM.

**d** The depletion of sodium bicarbonate from the mixed inorganic solution abolished the increase in intracellular calcium levels in MCF-GPR30 cells. **e, f** Sodium bicarbonate and potassium bicarbonate, but not potassium chloride, increased intracellular calcium levels in MCF-GPR30 (**e**) and HEK-GPR30 (**f**) cells in a dose-dependent manner. Statistical analysis: two-tailed unpaired t-test with Dunnett's correction (**a**) or Bonferroni's correction (**e** and **f**) after two-way ANOVA. Two-tailed unpaired t-test with Holm–Šídák's correction (**b**–**d**). Data are presented as mean values ± SEM. *P* values are shown if significant. ns indicates no significant difference. Source data are provided as a Source Data file.

that our $E_2$ preparation activated nuclear ERs. Other putative agonists for GPR30, including aldosterone (Fig. 1a) and G-1 (Supplementary Fig. 2c, d), also failed to activate GPR30 under our experimental conditions. However, the culture medium, Dulbecco's modified Eagle's medium (DMEM), induced elevation of intracellular calcium levels in MCF-GPR30 cells (Fig. 1b), which led us to explore GPR30-active components in DMEM. Neither amino acids, glucose (Supplementary Fig. 2e), vitamins and minerals (Supplementary Fig. 2f), nor their combination (Supplementary Fig. 2g) increased intracellular calcium in MCF-GPR30 cells. We assumed that one of the inorganic components of DMEM activated GPR30 and found that a mixed solution of inorganic salt components of DMEM elevated intracellular calcium levels in MCF-GPR30 cells (Supplementary Fig. 2h). This increase in intracellular calcium levels by the mixed inorganic solution was observed at various pH values (Fig. 1c). Minimum Essential Medium (MEM) also induced GPR30-dependent intracellular calcium level elevation (Fig. 1c). Depletion of sodium bicarbonate from the mixed inorganic solution abolished the intracellular calcium level elevation in MCF-GPR30 cells (Fig. 1d). Sodium bicarbonate and potassium bicarbonate, but not potassium chloride, induced an increase in intracellular calcium levels in MCF-GPR30 (Fig. 1e) and HEK293 cells stably expressing hGPR30 (HEK-GPR30) (Fig. 1f) in a dose-dependent manner. As the normal range of serum concentration of bicarbonate ions is 22–29 mM[27], these results substantiate that physiological concentrations of bicarbonate ions are responsible for calcium mobilisation through GPR30 activation. $E_2$ did not exert a synergistic, additive, or inhibitory effect on GPR30 activation induced by bicarbonate ions in the TGFα shedding assay (Supplementary Fig. 2i). Activation of GPR30 by bicarbonate ions was conserved among various species, including rats (Supplementary Fig. 2j), mice (Supplementary Fig. 2k), and zebrafish (Supplementary Fig. 2l).

## Bicarbonate triggers GPR30-Gq signalling

Next, we examined which subtypes of $G_\alpha$ proteins are coupled via GPR30 activation by bicarbonate ions. Previous reports have demonstrated $G_{\alpha q}$[28], $G_{\alpha i}$[17], and $G_{\alpha s}$[18]-coupled activation of GPR30. We performed a TGFα shedding assay using HEK293 cells devoid of $G_{\alpha q/11/12/13}$ and transfected with each indicated subtype of $G_\alpha$. The results showed that $G_{\alpha i1}$, $G_{\alpha i3}$, $G_{\alpha q}$, and $G_{\alpha 14}$ were promising candidates (Fig. 2a). Among these candidates, the $G_{\alpha i}$ family was not functionally coupled via the activation of GPR30 by bicarbonate ions because sodium bicarbonate did not reduce forskolin-induced cAMP production in HEK293 cells transiently expressing hGPR30 (Fig. 2b) or HEK-GPR30 (Fig. 2c). The increase in intracellular calcium levels induced by sodium bicarbonate was completely abolished by the $G_{\alpha q/11/14}$ inhibitor YM-254890 (Fig. 2d), but not by the pertussis toxin (PTX) that inhibits $G_{\alpha i}$ protein (Fig. 2d). The bicarbonate-GPR30 signal caused phosphorylation of ERK1/2 at Thr202/Tyr204, which was completely inhibited by YM-254890 (Fig. 2e), but not by PTX (Fig. 2f). Notably, the putative ligand $E_2$ did not induce phosphorylation of ERK1/2 under the same experimental conditions as those of sodium bicarbonate (Fig. 2g). The involvement of phospholipase C in the bicarbonate-GPR30 signal was also confirmed using the inositol phosphate accumulation assay; inositol phosphates were produced under activation of GPR30 by sodium bicarbonate, and its effective concentration ($EC_{50} = 11$–$12$ mM) was within the physiological range of bicarbonate concentration in vivo (Fig. 2h, i). These results indicate that the physiological concentration of bicarbonate ions induces signal transduction via GPR30 and $G_q$ proteins.

## Key amino acids of GPR30 in activation

We next investigated which amino acids of GPR30 are necessary for recognising bicarbonate ions. According to the public homology model (https://gpcrdb.org/)[29], many negatively charged amino acid residues are located deep inside the putative orthosteric pocket of

GPR30, where bicarbonate ions are unlikely to interact with amino acids. Therefore, we first surveyed positively charged residues on the extracellular edge of the putative orthosteric pocket (Fig. 3a). The conserved amino acids from humans to zebrafish, as zebrafish GPR30 was activated by bicarbonate ions (Supplementary Fig. 2l), include H200, H300, and H307 (Fig. 3b). H307 was selected as a potential amino acid residue for recognising bicarbonate ions because only the alanine substitution of H307 completely abolished the bicarbonate-induced activation of hGPR30 (Fig. 3c) and HA-tagged hGPR30 (Fig. 3d) in the TGFα shedding assay. Next, we selected seven hydrophilic amino acids located in the orthosteric pocket that potentially cooperate with H307 in the interaction with bicarbonate ions (Fig. 3e). A single alanine substitution of E115 or Q138 (Fig. 3f) completely abolished the bicarbonate-induced activation of hGPR30 (Fig. 3c) and HA-tagged hGPR30 (Fig. 3d). We confirmed a similar expression level of the HA-tagged hGPR30 mutants using western blotting (Fig. 3g, Supplementary Fig. 3a). Any substitution of these three amino acids with alanine nullified the bicarbonate-induced increase in intracellular calcium levels (Fig. 3h), with similar cell surface expression levels of GPR30 (Supplementary Fig. 3b). These results suggest that these three amino acids are essential for bicarbonate recognition and/or downstream signalling of GPR30.

We further performed a scintillation proximity assay (SPA) with the plasma membrane fraction of HEK293 cells (Supplementary Fig. 3c) transfected with wild-type GPR30-HA and the three mutants that were not activated by bicarbonate because of impaired bicarbonate recognition and/or downstream singling of GPR30. SPA using [14]C-labelled sodium bicarbonate showed that wild-type GPR30 caused a higher SPA count than mock (Supplementary Fig. 3d). As expected, the H307A mutant caused a significantly lower SPA count than wild-type GPR30-HA and the other mutants (Supplementary Fig. 3d). These results indicated that there was a specific association between bicarbonate ions and GPR30 and that H307 is likely to be involved in bicarbonate recognition, consistent with the results of selecting candidates for the recognition of bicarbonate ions (Fig. 3a–d).

## Bicarbonate-GPR30 signal in brain mural cells

Next, we examined whether endogenous GPR30 is activated by bicarbonate ions. In mouse myoblast C2C12 cells that endogenously express GPR30, sodium bicarbonate increased the intracellular calcium level in a concentration-dependent manner (Fig. 4a, b). Knockdown of *Gpr30* by stable expression of shRNA (knockdown efficiency: 87–90%, Fig. 4c) resulted in the amelioration of bicarbonate-induced increase in intracellular calcium level (Fig. 4d, Supplementary Fig. 4a, b).

Bicarbonate-induced activation of endogenous GPR30 in vitro prompted us to investigate the in vivo roles of the bicarbonate-GPR30 signal. To determine the type of cells that express GPR30 in vivo, we generated knock-in mice, in which the coding sequence of *Gpr30* was replaced with that of the fluorescent protein Venus (*Gpr30-Venus*-KI, Fig. 5a). In this mouse, *Venus* was expressed under the control of the *Gpr30* promoter. Confocal microscopic analyses of a heterozygous *Gpr30-Venus*-KI mouse (Supplementary Fig. 5a–e) showed strong expression of Venus in the brain (Supplementary Fig. 5a) and kidney (Supplementary Fig. 5d). Immunostaining of collagen type IV, which is expressed in the capillary basement membrane, followed by 3D reconstruction, revealed that Venus was expressed around the microvasculature in the brain (Fig. 5b). The brain microvasculature comprises endothelial cells, pericytes, and astrocytes, which cooperatively form the blood−brain barrier (BBB) and closely communicate with the neural system as the neurovascular unit (NVU)[13]. Higher expression levels of pericyte marker genes (*Pdgfrb* and *Abcc9*) and *Gpr30* were seen in Venus-positive microvascular cells (Venus[+]) isolated from heterozygous *Gpr30-Venus*-KI mouse brains than in Venus-negative (Venus[−]) microvascular cells (Fig. 5c). Endothelium-specific

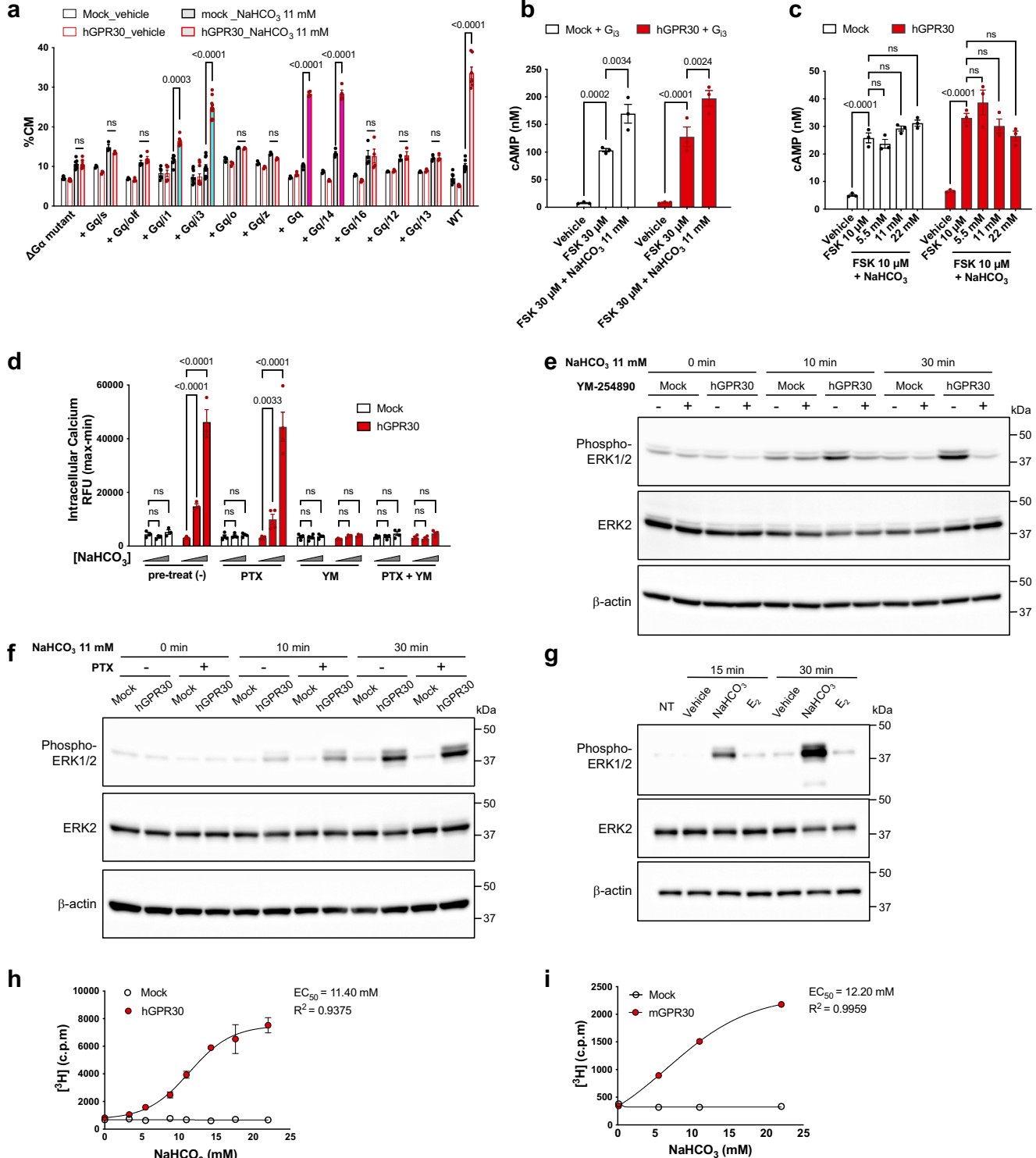

**Fig. 2 | Bicarbonate ions activate GPR30 signalling through the G$_q$ family of G proteins. a** TGFα shedding assay using HEK293 cells devoid of G$_{αq/11/12/13}$ and transfected with the indicated subtypes of G$_α$ proteins. Blue and magenta bars indicate G$_{αi}$ and G$_{αq}$ family candidates, respectively. **b, c** cAMP assay using HEK293 cells transiently (**b**) and stably (**c**) expressing hGPR30. Sodium bicarbonate did not inhibit forskolin (FSK)-dependent cAMP production in either cell line. **d** Calcium mobilisation assay using Fluo-8-loaded HEK-GPR30. The cells were treated with vehicle, 3.3 mM, and 11 mM NaHCO$_3$. Bicarbonate-induced intracellular calcium increase was completely abolished by the G$_{αq/11/14}$ inhibitor YM-254890 (YM, 1 μM, 45 min) but not by pertussis toxin (PTX, 100 ng/ml, 16 h). **e, f** HEK293 cells transiently expressing hGPR30 were pretreated with YM-254890 (**e**) or PTX (**f**) and then treated with 11 mM NaHCO$_3$ for the indicated periods. β-actin served as the loading

control. **g** GPR30-dependent phosphorylation of ERK1/2 was evaluated using western blotting. HEK293 cells transiently expressing hGPR30 were treated with vehicle, 11 mM NaHCO$_3$, or 100 nM E$_2$ and harvested at 15 and 30 min after stimulation. β-actin serves as a loading control. **h, i** GPR30-dependent accumulation of inositol phosphates. HEK293 cells transiently expressing hGPR30 (**h**) and MCF-mGPR30 cells (**i**) were treated with indicated sodium bicarbonate concentrations. Nonlinear regression (four parameters) was performed. The EC$_{50}$ values of sodium bicarbonate were 11.40 mM (**h**) and 12.20 mM (**i**). Statistical analysis: two-tailed unpaired t-test with Bonferroni's correction after two-way ANOVA (**a**–**d**). Data are presented as mean values ± SEM. *P* values are shown if significant. ns indicates no significant difference. Source data are provided as a Source Data file.

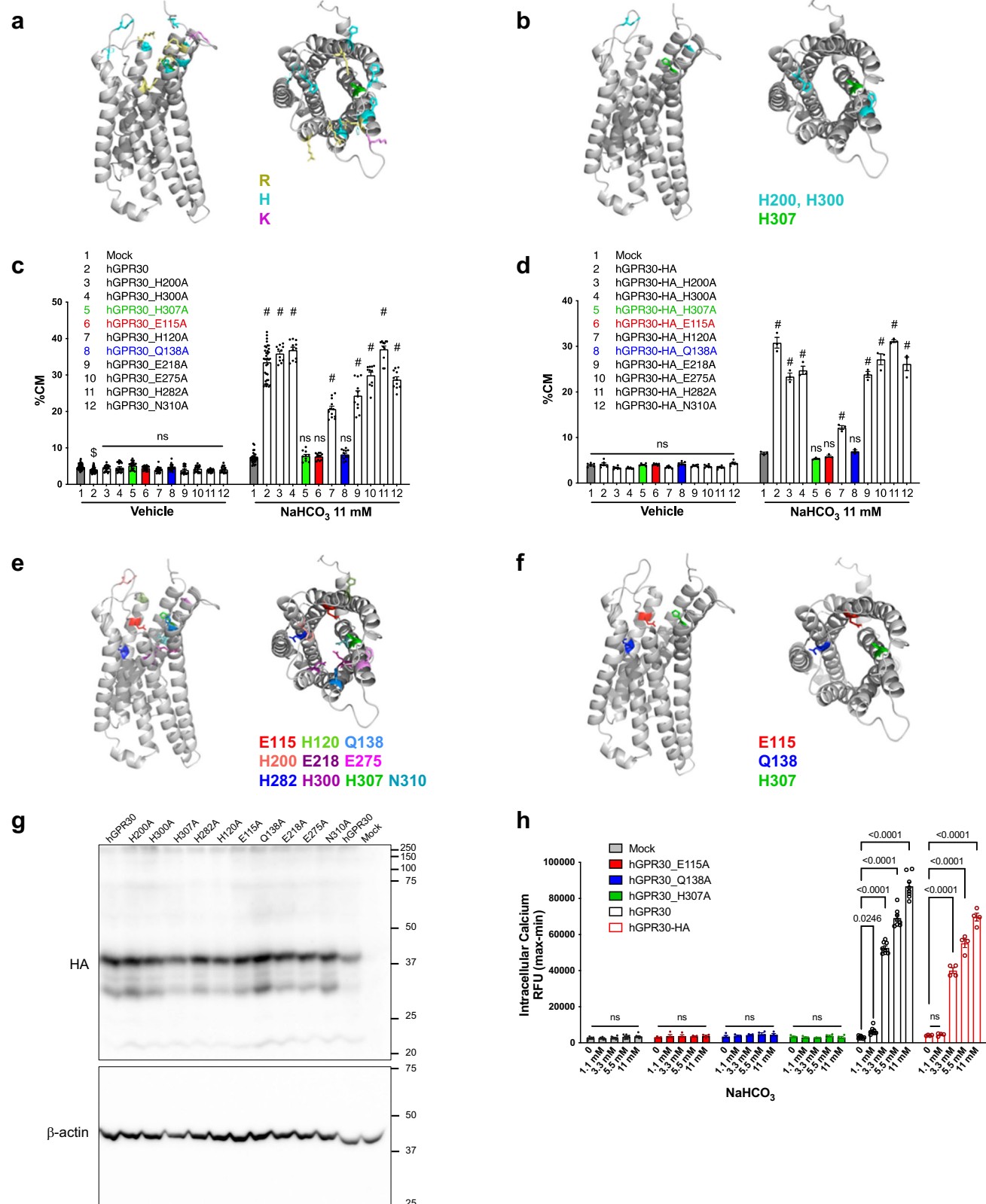

genes (*Cldn5* and *Tie2*) were not enriched in the Venus+ cells (Fig. 5d). In situ hybridisation analysis demonstrated that *Gpr30* mRNA was detected around the large vessels and microvasculature in the cortex, cerebellum, and hippocampus (Fig. 5e). Multiple in situ hybridisation analyses showed that *Gpr30* was expressed almost exclusively in mural cells in the brain cortex, which included *Acta2+ Pdgfrb+* vascular

smooth muscle cells (SMCs) and *Acta2- Pdgfrb+* pericytes (Fig. 5f, g). Fluorescence-activated cell sorting (FACS) analysis also showed that pericytes (CD41-CD45-CD31-CD13+)[30], but not endothelial cells (CD41-CD45-CD31+CD13-)[30], were distinguished by the expression of *Gpr30* (Fig. 5h, i). These results clarified that GPR30 is expressed in brain mural cells at the transcriptional level, which is consistent with a

**Fig. 3 | Identification of amino acid residues essential for GPR30 activation by bicarbonate. a, b** The predicted model of human GRP30 was generated based on the active conformation of CC chemokine receptor 5 (PDB:7O7 F, https://www.rcsb.org/structure/7O7F) in combination with other similar coordinates obtained from GPCRdb (https://gpcrdb.org/). The candidate residues for bicarbonate coordination were selected based on their side chain properties (e.g. positively charged residues: Arg (R), His (H), and Lys (K)) (**a**) and their conservation across animals (**b**) and mapped on the model structure. **c, d** TGFα shedding assay using HEK293 cells transfected with hGPR30 (**c**) or HA-tagged hGPR30 (**d**). The mutants H307A, E115A, and Q138A are highlighted in green, red, and blue, respectively. **e** The candidate residues analysed in our study were mapped on the model structure. **f** Three

residues identified as important for bicarbonate-dependent activation were mapped on the model structure. **g** Western blotting analysis for HA-tagged hGPR30 mutants (upper panel) and β-actin control (lower panel). **h** Calcium mobilisation assay using HEK293 cells stably expressing three mutants (E115A, Q139A, and H307A) of hGPR30. Statistical analysis: $^{§}p < 0.05$, $^{#}p < 0.0001$ compared to mock cells using two-tailed unpaired t-test with Bonferroni's correction after two-way ANOVA (**c, d**). Two-tailed unpaired t-test with Bonferroni's correction after two-way ANOVA (**h**). Data are presented as mean values ± SEM. *P* values are shown if significant. ns indicates no significant difference. Source data are provided as a Source Data file.

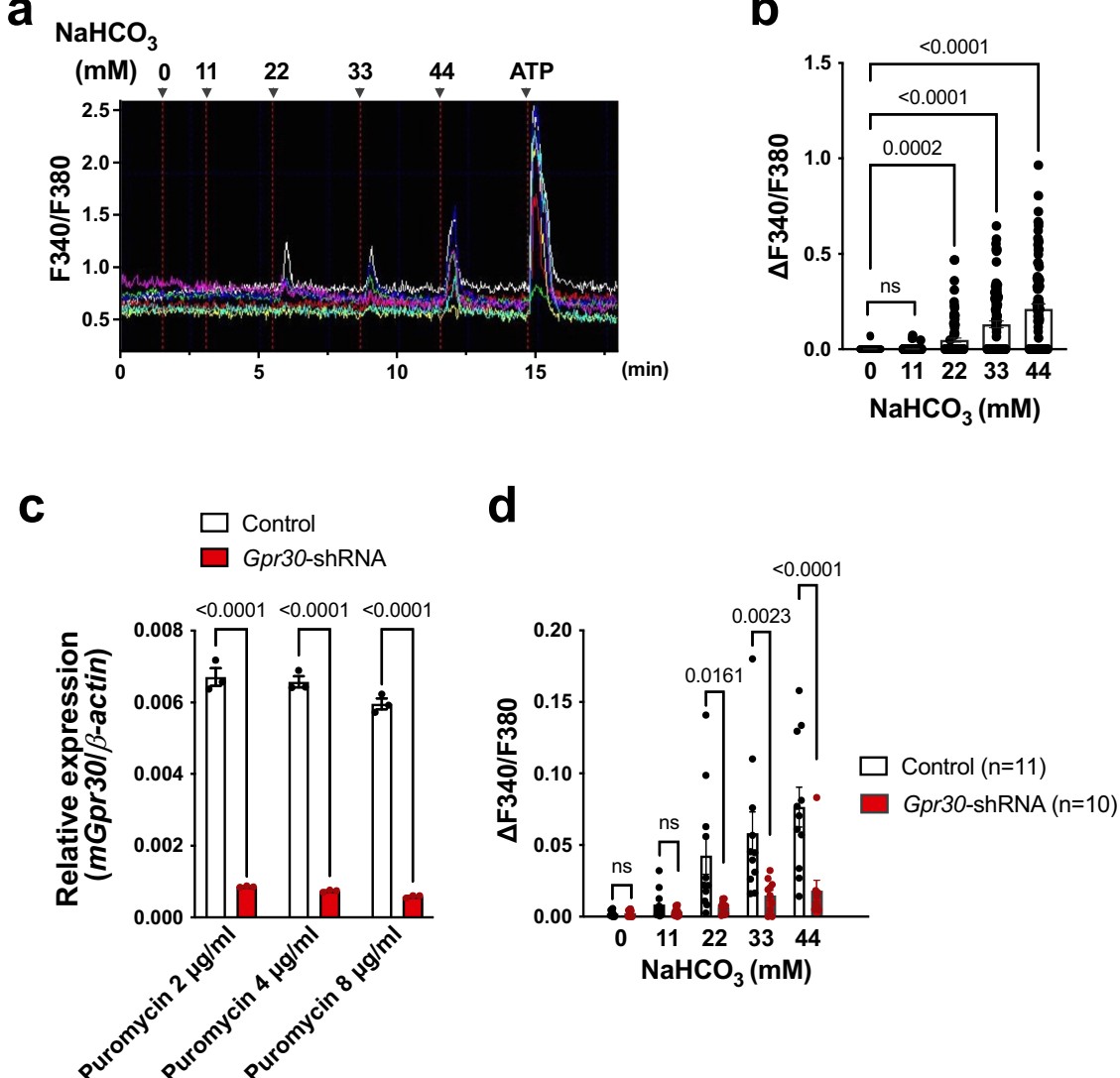

**Fig. 4 | Endogenous GPR30 is activated by bicarbonate ions.** Calcium mobilisation by sodium bicarbonate in Fura-2-loaded mouse myoblast C2C12 endogenously expressing GPR30. **a, b** C2C12 cells (*n* = 89 cells in one experiment) were stimulated with the indicated concentrations of sodium bicarbonate and 100 μM ATP. Representative image (**a**) and quantification (**b**). **c** Knockdown efficiency (87–90%) of *Gpr30* in puromycin (2–8 μg/ml)-resistant stable C2C12 cells was assessed using quantitative RT-PCR. **d** Puromycin (4 μg/ml)-resistant C2C12 cells

stably expressing the control or *Gpr30*-shRNA were stimulated with the indicated concentrations of sodium bicarbonate. Statistical analysis: two-tailed paired t-test with Bonferroni's correction after repeated measures one-way ANOVA (**b**) or two-tailed unpaired t-test with Bonferroni's correction after two-way ANOVA (**c, d**). Data are presented as mean values ± SEM. *P* values are shown if significant. ns indicates no significant difference. Source data are provided as a Source Data file.

previous knock-in experiment[31] and recent single-cell RNA-sequencing analyses[15,16,32,33], demonstrating the expression of GPR30 in brain mural cells, including pericytes and SMCs.

To clarify the bicarbonate-GPR30 signal in brain pericytes, we established a primary culture of the brain microvascular fraction

and performed an intracellular calcium mobilisation assay using *Gpr30*-positive and -negative pericytes harvested from *Gpr30*[+/Venus] and *Gpr30*[Venus/Venus] mice, respectively. *Gpr30*-positive pericytes (*Gpr30*[+/Venus]) showed a rapid increase in intracellular calcium levels owing to the physiological concentration of bicarbonate ions

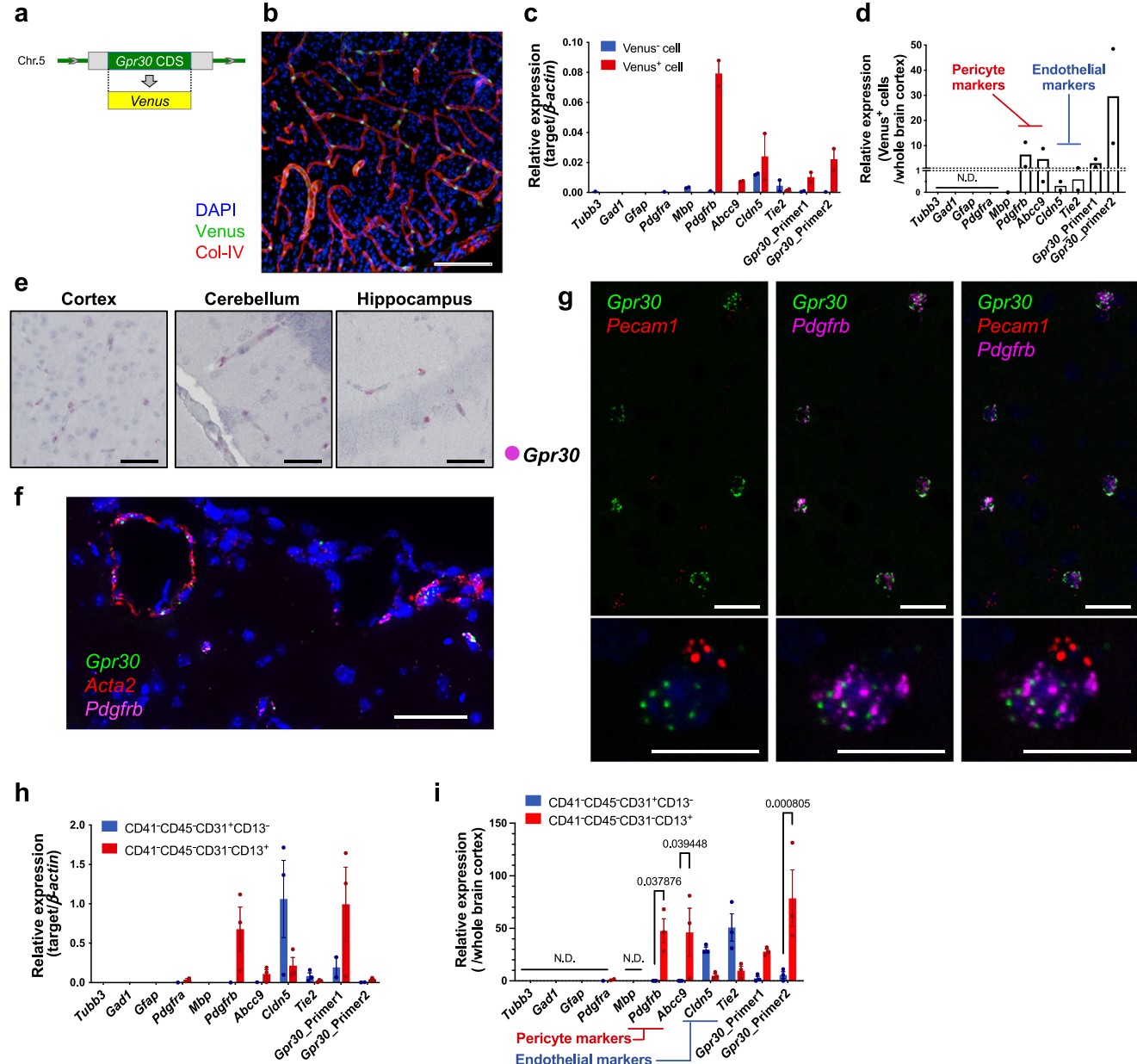

**Fig. 5 | GPR30 is expressed in brain mural cells. a** Design of the *Gpr30-Venus*-KI construct. The coding sequence of *Gpr30* was replaced in frame with that of *Venus*. **b** Confocal microscopy image of a heterozygous *Gpr30-Venus*-KI (*Gpr30^+/Venus^*) mouse brain. Slices (50 μm) of the brains were stained with an antibody for collagen type IV (Col-IV) and DAPI. Scale bars, 100 μm. **c, d** Quantitative RT-PCR analyses of marker genes for neurons, astrocytes, oligodendrocytes, oligodendrocyte progenitor cells, pericytes, and endothelial cells. Relative expression levels of the genes in Venus⁺ cells compared with those in the whole brain cortex are shown in (**d**). **e** In situ hybridisation analyses of brain sections from wild-type (*Gpr30^+/+^*) mice. *Gpr30* mRNA is visualised as magenta dots. Scale bars, 50 μm. **f** Visualisation of *Gpr30* (green), the vascular smooth muscle cell (SMC) marker *Acta2* (red), and the mural cell marker *Pdgfrb* (magenta) using multiple in situ hybridisation. Scale bar, 50 μm. **g** Visualisation of *Gpr30* (green), endothelial marker *Pecam1* (red), and pericyte marker *Pdgfrb* (magenta) using multiple in situ hybridisation. Scale bars, 10 μm. **h, i** Quantitative RT-PCR analyses of the marker genes for neurons, astrocytes, oligodendrocytes, oligodendrocyte progenitor cells, pericytes, and endothelial cells. CD41⁻CD45⁻CD31⁺CD13⁻ and CD41⁻CD45⁻CD31⁻CD13⁺ microvascular cells were isolated from the brain cortex of *Gpr30^+/+^* mice. Relative expression levels of the genes compared with those in the whole brain cortex are shown in (**i**). Statistical analysis: two-tailed unpaired t-test with Holm-Šídák's correction (**i**). Data are presented as mean values ± SEM. *P* values are shown if significant. N.D. indicates not detected. Source data are provided as a Source Data file.

(Fig. 6a), whereas *Gpr30*-negative pericytes (*Gpr30^Venus/Venus^*) did not (Fig. 6b, c).

As phenotypic drifts may alter the responses in primary cultures, we further performed calcium imaging in freshly isolated SMCs and pericytes independently from the mice in which GCaMP expression was induced by codon-improved Cre recombinase (iCre) that was driven by the *Gpr30* promoter (Supplementary Fig. 6a–d). We found GPR30-dependent, bicarbonate-induced increases in calcium levels in SMCs (Fig. 6d–f) and pericytes (Fig. 6g–i). These

results demonstrate that endogenous GPR30 in both SMCs and pericytes senses bicarbonate to activate intracellular signalling cascades. Collectively, these results indicate that GPR30, endogenously expressed in brain mural cells, is a functional bicarbonate receptor.

## GPR30 contributes to ischaemia-reperfusion injury
We aimed to explore the functional significance of GPR30 deficiency in the cerebral microvasculature using two lines of genetically modified

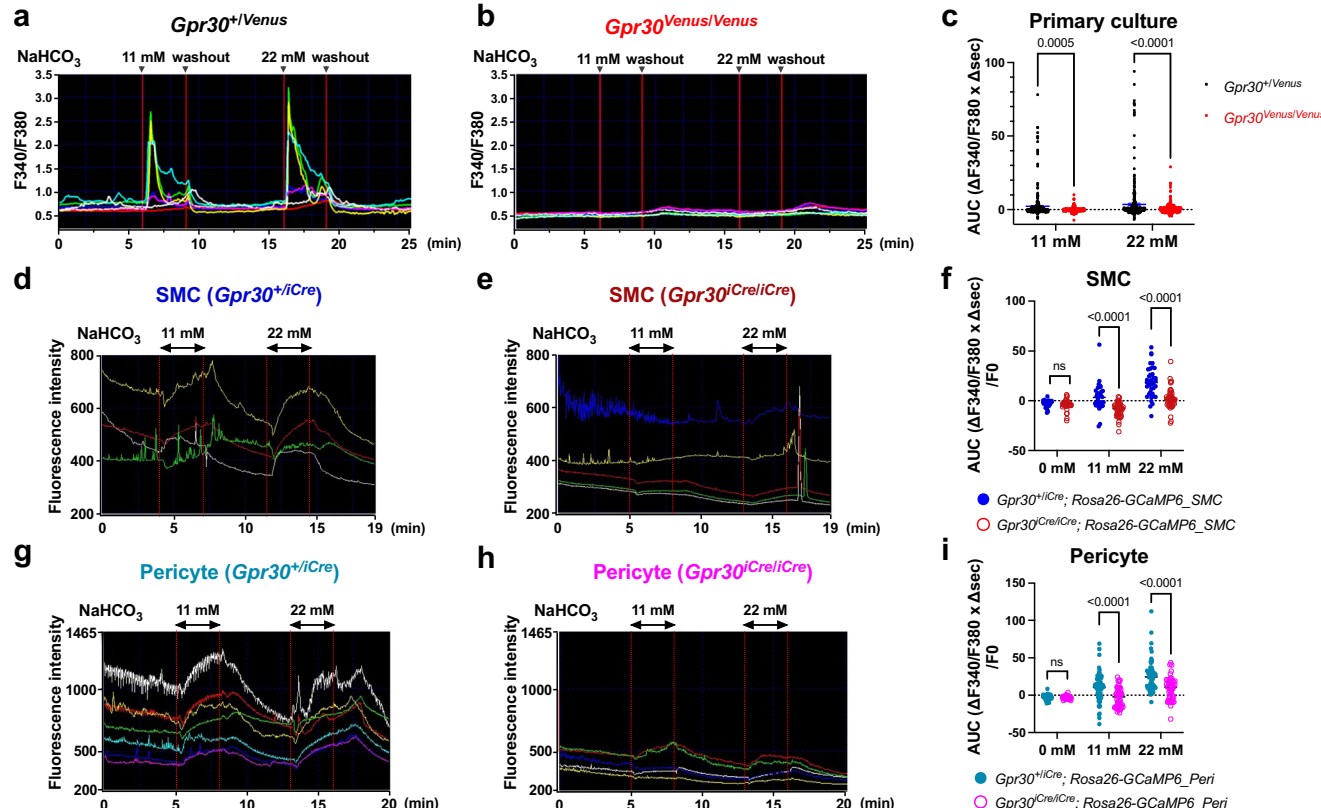

**Fig. 6 | GPR30 expressed in brain mural cells is activated by bicarbonate.**
**a**–**c** Calcium mobilisation assay for endogenous GPR30 in primary culture. The primary culture of Venus-positive cells harvested from the brain cortex of *Gpr30*-heterozygous (**a**, *Gpr30*$^{+/Venus}$) and *Gpr30*-null (**b**, *Gpr30*$^{Venus/Venus}$) mice were subjected to calcium imaging. **c** Quantification of a) and b) as area under the curve. **d**–**i** Calcium mobilisation assay for freshly isolated SMCs (**d**–**f**) and pericytes (**g**–**i**). Freshly isolated mCherry-positive cells harvested from the cerebral arteries (**d**, **e**)

and cortex (**g**, **h**) of *Gpr30*-heterozygous (**d**, **g**, *Gpr30*$^{+/iCre}$; *Rosa26-GCaMP6*) and *Gpr30*-null (**e**, **h**, *Gpr30*$^{iCre/iCre}$; *Rosa26-GCaMP6*) mice were subjected to calcium imaging. **f**, **i** Quantification of **d** and **e**, **g** and **h**, as area under the curve, respectively. Statistical analysis: two-tailed unpaired t-test with Bonferroni's correction after two-way ANOVA (**c**, **f**, **i**). Data are presented as mean values ± SEM. *P* values are shown if significant. ns indicates no significant difference. Source data are provided as a Source Data file.

mice (*Gpr30*-knockout line: *Gpr30*$^{+/+}$ and *Gpr30*$^{-/-}$, *Gpr30*-*Venus*-KI line: *Gpr30*$^{+/Venus}$, *Gpr30*$^{-/Venus}$, and *Gpr30*$^{Venus/Venus}$). At steady state, GPR30-deficient mice were healthy and showed no growth abnormalities. The expression of neuronal, astrocytic, and synaptic markers in the brain cortex (Supplementary Fig. 7a) and hippocampus (Supplementary Fig. 7b) was comparable among all genotypes, indicating no obvious defects in neural maturation and function in GPR30-deficient mice.

Heterozygous *Gpr30*-*Venus*-KI mice (*Gpr30*$^{+/Venus}$ and *Gpr30*$^{-/Venus}$), which are expected to express similar levels of Venus, were used for the analysis of pericyte/endothelial coverage of the brain microvasculature. *Gpr30*$^{-/Venus}$ mice were functionally equivalent to *Gpr30*$^{Venus/Venus}$ and *Gpr30*$^{-/-}$ mice. Pericyte coverage of blood vessels was almost equivalent (Supplementary Fig. 7c), although with slightly higher endothelial coverage in *Gpr30*$^{-/Venus}$ mice than in *Gpr30*$^{+/Venus}$ mice (Supplementary Fig. 7d). No morphological abnormalities were detected using electron microscopy (Supplementary Fig. 7e). Vascular permeability evaluated via the leakage of tracers with different molecular weights was not influenced by GPR30 deficiency (Supplementary Fig. 7f–h). Overall, these results indicate that GPR30 deficiency does not affect BBB integrity at steady state.

Brain pericytes have been reported to play pivotal pathophysiological roles in cerebral ischaemia-reperfusion injury[34]. Therefore, we subjected GPR30-deficient mice to a transient middle cerebral artery occlusion (MCAO), one of the most frequently utilised models to study cerebral ischaemia-reperfusion injury (Fig. 7a). Because previous studies suggested that premenopausal females were

protected against capillary dysfunction and brain injury under particular conditions[35], we subjected only male mice to MCAO to evaluate the effects of GPR30 in ischaemia-reperfusion injury. Milder ischaemia-reperfusion injury was seen in two lines of GPR30-deficient mice (*Gpr30*$^{Venus/Venus}$ and *Gpr30*$^{-/-}$) than in *Gpr30*$^{+/Venus}$ and *Gpr30*$^{+/+}$ mice, respectively (Fig. 7, Supplementary Fig. 8). Neurological deficits, as shown by a higher modified Neurological Severity Score (mNSS), were reduced in *Gpr30*$^{Venus/Venus}$ (Fig. 7b) and *Gpr30*$^{-/-}$ (Supplementary Fig. 8a) mice. Milder BBB disruption (Fig. 7c, d, Supplementary Fig. 8b, c) and smaller infarct volumes (Fig. 7e, f, Supplementary Fig. 8d, e) were observed 3 and 7 days after reperfusion in *Gpr30*$^{Venus/Venus}$ and *Gpr30*$^{-/-}$ mice than in *Gpr30*$^{+/Venus}$ and *Gpr30*$^{+/+}$ mice, respectively. In contrast to the numerous terminal deoxynucleotidyl transferase dUTP nick end labelling (TUNEL)-positive apoptotic cells in *Gpr30*$^{+/Venus}$ and *Gpr30*$^{+/+}$ brains, apoptotic cells were hardly detectable in GPR30-deficient mice 3 days after reperfusion (Fig. 7g, h, Supplementary Fig. 8f, g). These results demonstrate that GPR30 deficiency protects mice from ischaemia-reperfusion injury.

### GPR30 controls cerebral reperfusion
We then investigated how GPR30-deficient mice were protected from ischaemia-reperfusion injury. *Gpr30*$^{Venus/Venus}$ mice exhibited milder neurological impairments than *Gpr30*$^{+/Venus}$ mice as early as 1 h after MCAO (Fig. 7b). This indicates that GPR30 deficiency affects the pathophysiology of ischaemia-reperfusion injury during the hyperacute and acute phases (<24 h), rather than during the subacute phase

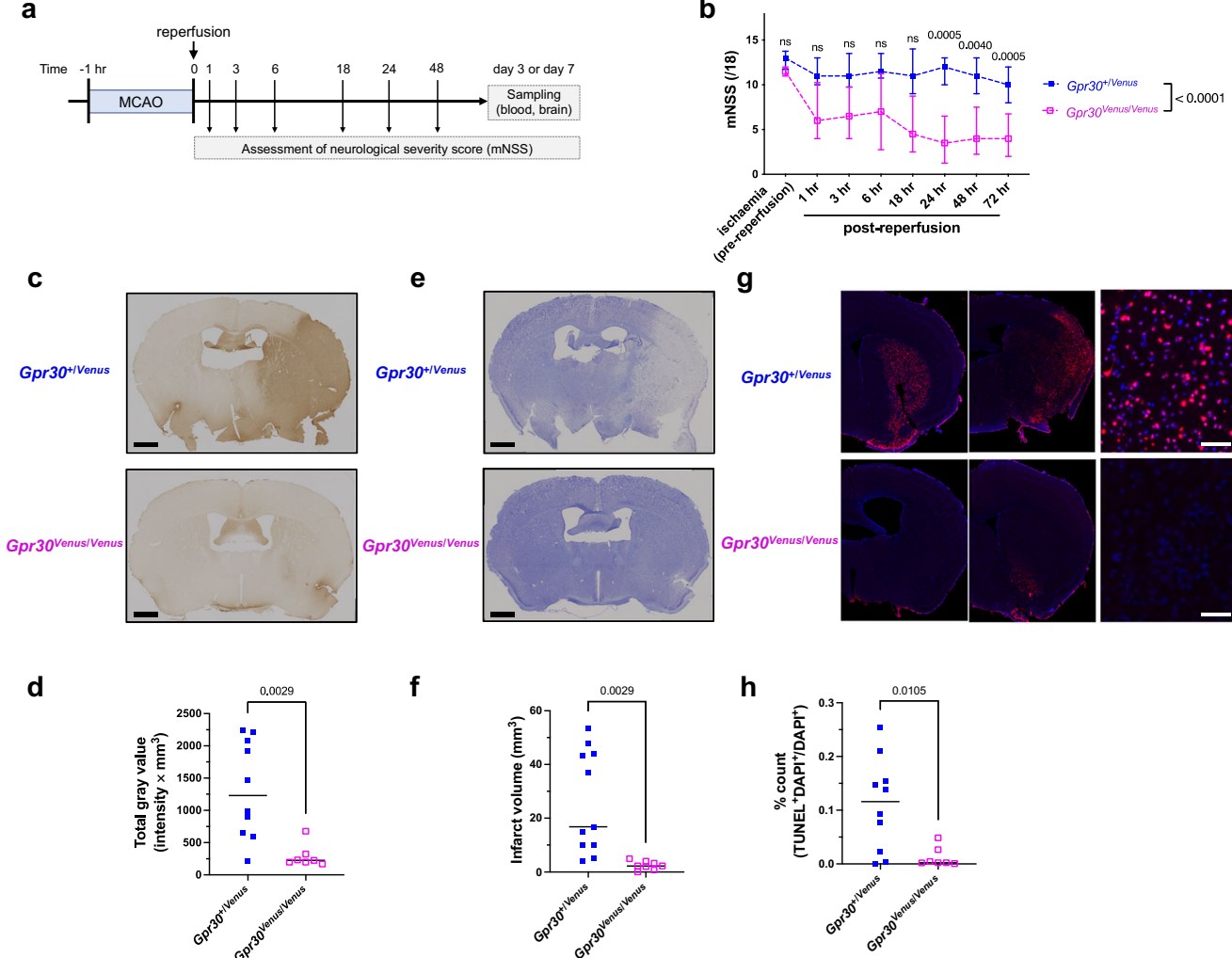

**Fig. 7 | GPR30 deficiency protects against ischaemia-reperfusion injury after transient middle cerebral artery occlusion. a** Experimental design of a transient middle cerebral artery occlusion (MCAO) model. **b** Modified Neurological Severity Score (mNSS) evaluated over time after MCAO. **c, d** Evaluation of the blood–brain barrier disruption. Representative images (**c**) and quantification (**d**) of IgG staining of whole brain sections 3 days after MCAO. Scale bars, 1 mm. **e, f** Evaluation of infarct volume. Representative images (**e**) and quantification (**f**) of cresyl violet staining of whole brain sections 3 days after MCAO. Scale bars, 1 mm. **g, h** Evaluation of apoptosis using TUNEL staining of whole brain sections 3 days after MCAO. Representative images (**g**) and quantification (**h**) of hemibrain sections 3 days after MCAO. The ratio of TUNEL-positive nuclei to total nuclei was calculated on hemibrain sections. TUNEL-positive apoptotic cells were scarcely detectable in GPR30-deficient (*Gpr30*<sup>Venus/Venus</sup>) mice. Scale bars, 50 μm. Statistical analysis: **b** two-tailed mixed-effects analysis with Bonferroni's multiple comparison correction. Data are presented as the median ± interquartile range. **d, f, h** two-tailed unpaired t-test. Data are presented as dot plots with the median. *P* values are shown if significant. ns indicates no significant difference. Source data are provided as a Source Data file.

(-3 days), when differentiation and migration of pericytes are reported to contribute to tissue remodelling after MCAO[36].

As previous reports have demonstrated that ischaemia and reperfusion cause rapid changes in blood gas values, electrolytes, and glucose[37], we examined whether bicarbonate concentrations shifted between ischaemia and reperfusion. We measured the concentration of bicarbonate ions in the blood collected from the facial vein, which directly connects to the cavernous sinus (Fig. 8a). Successful occlusion of the middle cerebral artery (MCA) was confirmed by higher mNSS scores (Supplementary Fig. 9a). The concentration of bicarbonate ions in the serum was increased within 5 min of reperfusion regardless of the *Gpr30* genotype, whereas it remained unchanged in sham-operated mice (Fig. 8b). The concentrations of other electrolytes (Ca²⁺, Mg²⁺, Na⁺, Cl⁻, and inorganic phosphorus [IP]), were not significantly altered by reperfusion (Supplementary Fig. 9b–f). Serum bicarbonate levels were comparable among *Gpr30*<sup>+/+</sup>, *Gpr30*<sup>-/-</sup>, *Gpr30*<sup>+/Venus</sup>, and *Gpr30*<sup>Venus/Venus</sup> mice at steady state (Supplementary Fig. 10a) and at 3 (Supplementary Fig. 10b) and 7 days (Supplementary Fig. 10c) after MCAO.

GPR30 is expressed in mural cells, vascular SMCs of large vessels such as arteries and arterioles, and pericytes of capillaries, both of which are contractile and control blood flow[38,39]. We hypothesised that bicarbonate ions induce pericyte and/or SMC contraction through GPR30, contributing to prolonged local hypoperfusion due to capillary constriction. To test this hypothesis and to clarify which type of mural cells contribute to protection from ischaemia-reperfusion injury in GPR30-deficient mice, we performed magnetic resonance angiography (MRA) and laser Doppler flowmetry to analyse the blood flow in large vessels and capillaries, respectively.

MRA was performed before and 30 min after reperfusion in *Gpr30*<sup>+/+</sup>, *Gpr30*<sup>+/Venus</sup>, and GPR30-deficient (*Gpr30*<sup>-/-</sup> and *Gpr30*<sup>Venus/Venus</sup>) mice (Fig. 8c). Widespread cerebral ischaemia over the left MCA region was induced by its occlusion, regardless of the *Gpr30* genotype (Fig. 8d, top). MRA 30 min after reperfusion, however, revealed that GPR30-deficient mice exhibited increased recovery of blood flow in the left MCA region compared to that in *Gpr30*<sup>+/+</sup> and *Gpr30*<sup>+/Venus</sup> mice (Fig. 8d, middle and bottom, Fig. 8e).

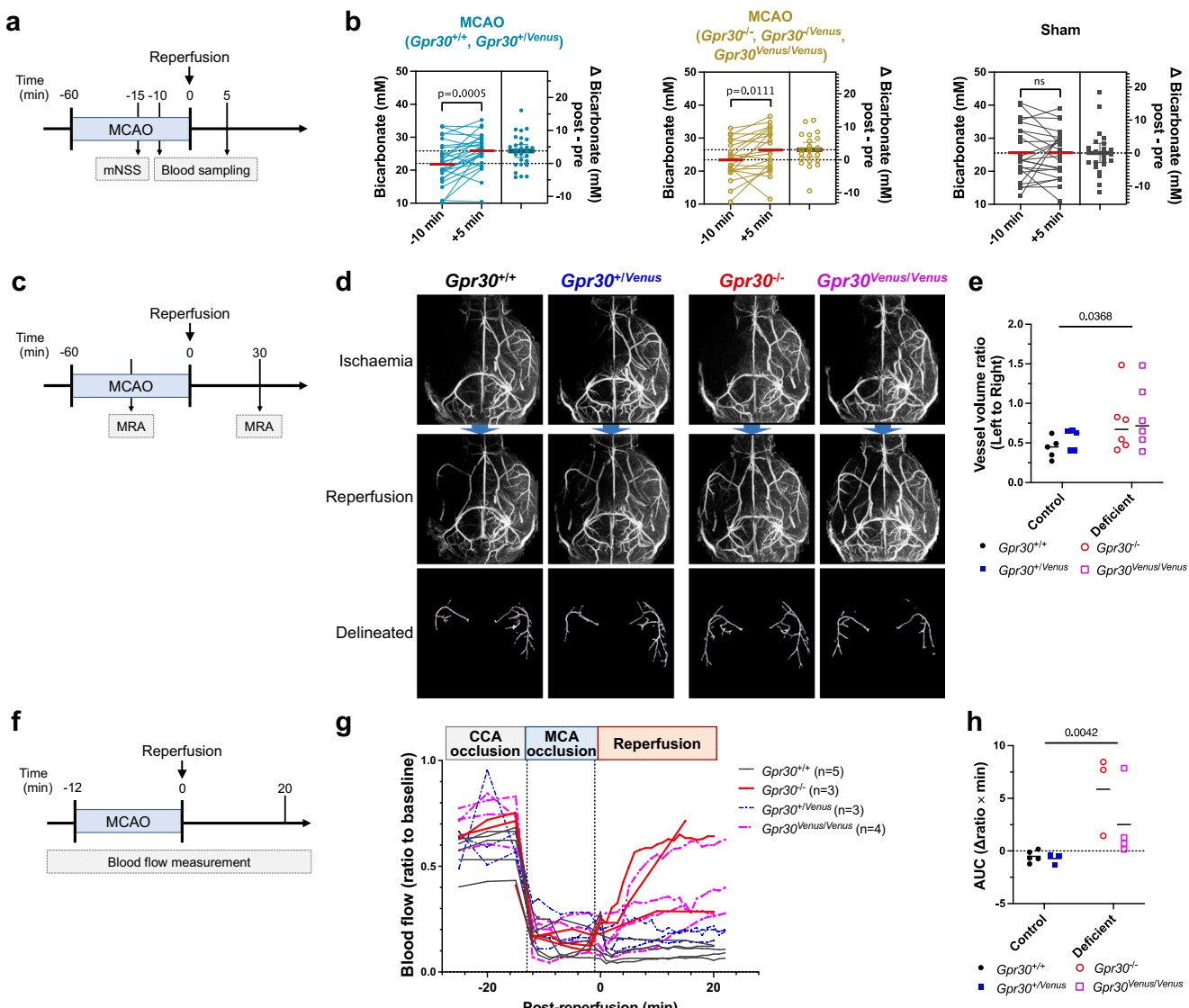

**Fig. 8 | Rapid recovery of cerebral blood flow in the MCA region upon reperfusion in Gpr30-deficient mice. a, b** Measurement of serum concentration of bicarbonate ions during ischaemia and reperfusion. **a** Experimental design. The modified Neurological Severity Score (mNSS) was evaluated at 45 min of MCAO, and blood was collected before (−10 min) and after (+5 min) reperfusion. **b** Changes in serum bicarbonate levels in *Gpr30+/+* and *Gpr30+/Venus* (left), *Gpr30−/−*, *Gpr30−/Venus*, and *Gpr30Venus/Venus* (middle), and sham-operated (right) mice. A set of data collected at pre- and post-reperfusion from the same mouse are connected with a line (left box). The difference in serum bicarbonate concentration (Δbicarbonate (mM) post − pre) presented as dot plots with mean (right box). **c−e** Magnetic resonance angiography (MRA) analysis before and after reperfusion.

**c** Experimental design. **d** Maximal intensity projection images before (top) and after (middle) reperfusion. Delineated MCA and branches after reperfusion are shown (bottom). **e** Vessel volume of the left MCA and its branches (reperfusion side) as a ratio to that of the right (control side). **f−h** Measurement of cerebral blood flow using laser Doppler flowmetry. **f** Experimental design. **g** Normalised blood flow as a ratio to the basal blood flow. **h** Area under the curve of the blood flow from 0 to 20 min during the reperfusion phase. Statistical analysis: **b** two-tailed paired t-test. **e**, **h** the main effect in two-way ANOVA. Data are presented as dot plots with median. *P* values are shown if significant. ns indicates no significant difference. Source data are provided as a Source Data file.

Microcirculation was continuously monitored using laser Doppler flowmetry in mice with MCAO (Fig. 8f). The cerebral blood flow in the left MCA region was reduced to a comparable level during the MCA occlusion phase in all mice (Fig. 8g). *Gpr30+/+* and *Gpr30+/Venus* mice showed prolonged hypoperfusion even after recanalisation (Fig. 8g, reperfusion phase). By contrast, GPR30-deficient mice demonstrated recovered cerebral blood flow more rapidly and to a higher level. A significant increase in blood flow recovery was observed during the first 20 min of the reperfusion phase in GPR30-deficient mice compared with that in *Gpr30+/+* and *Gpr30+/Venus* mice (Fig. 8h). Collectively, it is likely that GPR30 expressed in SMCs and pericytes attenuates blood flow recovery after reperfusion.

## Discussion

In this study, we searched for an acid−base balance-related GPCR and focused on GPR30 to study its contribution to the pathophysiology of ischaemic stroke. The subsequent serendipitous finding of GPR30 activation by the culture medium led to the identification of GPR30 as a bicarbonate-sensing receptor. We further revealed that GPR30 expressed in the brain mural cells reacted to bicarbonate and that GPR30-deficient mice were protected from ischaemia-reperfusion injury by rapid blood flow recovery. Together, we identified GPR30 as a bicarbonate-sensing GPCR that underlies the pathophysiology of ischaemic stroke.

Because protons and bicarbonate shift concomitantly to equilibrium in the bicarbonate buffering system, it is difficult to distinguish

bicarbonate-mediated cellular responses from pH-dependent reactions. However, several studies have demonstrated that bicarbonate itself exacerbates ischaemia-reperfusion injury independently of pH[40,41], raising the possibility of a bicarbonate-sensing machinery and the related downstream signalling pathway. Our study unraveled this previously unanswered question in that we identified GPR30 as a pH-independent, bicarbonate-sensing GPCR that underlies ischaemia-reperfusion injury. It should be noted, however, that soluble adenylyl cyclase (sAC) is another evolutionarily conserved bicarbonate-sensing machinery in sperm[42].

GPR30 was initially identified to be widely expressed in the brain and peripheral tissues[43], but high expression levels of GPR30 in ER-positive breast cancer cell line[44] and a non-genomic action of $E_2$ via GPR30[45] led to the recognition of GPR30 as an oestrogen-binding receptor[17,18]. The tissue specificity of *Gpr30* expression (Supplementary Fig. 1b) and our results (Fig. 5b, e, Supplementary Fig. 5a–e) support the initial reports of the wide distribution of the receptor in the brain and peripheral tissues. Furthermore, the physiological concentration of bicarbonate activated GPR30 in various animal species, whereas the physiological concentration (100–1,000 pM) of $E_2$ did not. These results substantiate that GPR30 intrinsically functions as a bicarbonate-sensing system in cerebral microcirculation, although the activation of GPR30 by bicarbonate and $E_2$ is not mutually exclusive.

One possible question arising from these data is whether bicarbonate ions are a ligand or a modulator for GPR30, considering a recent report on the allosteric effect of bicarbonate[46]. Our study showed that bicarbonate itself activated GPR30, both in the presence and absence of $E_2$ (Supplementary Fig. 2i). Despite the possibility of unrecognised molecules activating GPR30, it is most likely that bicarbonate is a natural ligand for GPR30, at least until another molecule, if any, achieves consensus as a genuine ligand.

We identified three amino acids (E115, Q138, and H307) in GPR30 that are crucial for the bicarbonate-GPR30 signal. H307 and E115 are rather close, whereas Q138 is more than 10 Å from the other two residues. Therefore, it is unlikely that all three residues directly coordinate bicarbonate ions together. Because E115A and Q138A mutants caused a SPA count that was as high as that of wild-type GPR30-HA (Supplementary Fig. 3d), E115 and Q138 are unlikely to be involved in bicarbonate recognition. It is possible that binding of the bicarbonate ion to H307 induces rearrangement of these residues and leads to receptor activation. Future structural analyses of the complex of GPR30 and bicarbonate ions using X-ray crystallography or cryo-EM will address this issue.

We acknowledge a few limitations in our in vivo study. First, while we hypothesised that sudden haemodynamic changes during ischaemia and reperfusion lead to a rapid change in the local bicarbonate concentration, activating GPR30, we only detected slight but significant shifts in serum bicarbonate levels from 22 mM during ischaemia to 26 mM after reperfusion (Fig. 8b). These shifts were out of activation range of GPR30 signalling as indicated by the concentration-response curves in our in vitro experiments (Fig. 2h, i). The change in serum bicarbonate levels (Fig. 8b) might not precisely mirror the change in local bicarbonate levels in the brain because the systemic bicarbonate buffering system attenuates local dynamic shifts. Previous studies consistently demonstrated that ischaemia causes a significant change in the local acid–base balance in the brain, i.e., a decrease in pH values (from 7.3 to 6.0–6.5), an increase in $pCO_2$ (from 40–50 to >100 mmHg), and a decrease in the $HCO_3^-$ concentrations (from 20–26 to 11–12 mM extracellularly), which are restored to the pre-ischaemic state by reperfusion[47–49]. Contrastingly, arterial and venous pH values, $pCO_2$, and $HCO_3^-$ concentrations showed only slight shifts during ischaemia and reperfusion[48,49]. Therefore, in our experiment, it is speculated that GPR30 was deactivated due to a decrease in local bicarbonate concentrations from its physiological level (22–26 mM) to the $EC_{50}$ level (11–12 mM) during ischaemia and reactivated due to a

recovery in local bicarbonate concentrations enough for full activation (26–27 mM) during transitions from ischaemia to reperfusion.

A restricted blood flow limits the supply of oxygen to tissues. Accordingly, the cellular metabolic state shifts from aerobic respiration to anaerobic glycolysis, producing lactate that causes intracellular acidosis. Mechanisms such as the export of lactate via monocarboxylate transporters, activation of the $Na^+/H^+$ exchanger, and import of bicarbonate ions via SLC4A family transporters[50], counteract intracellular acidosis and cause extracellular acidosis. All these mechanisms contribute to the reduction of extracellular bicarbonate levels[51]. Ischaemia and reperfusion also cause significant changes in ion homoeostasis, as related to sodium, potassium, ionised calcium, chloride, and glucose[52,53]. Although local changes in pH values, electrolyte levels, and glucose levels are possibly involved in the pathophysiology of ischaemia-reperfusion injury, none of these variances, other than bicarbonate ions, activated GPR30 in vitro (Fig. 1d, Supplementary Fig. 2e).

Collectively, we assume that shifts in local bicarbonate levels during ischaemia and reperfusion affected GPR30 activation in vivo, although we cannot exclude the possibilities that the constitutive activation of GPR30 throughout ischaemia and reperfusion participates in the pathophysiology of ischaemia-reperfusion injury and that the dynamic range of GPR30 activation by bicarbonate in vivo differs from that in vitro. The direct measurement of dynamic shifts in local bicarbonate levels would be a topic for future research for verifying bicarbonate-GPR30 signalling during ischemia–reperfusion.

Second, although we demonstrated that SMCs and pericytes responded to bicarbonate ex vivo and that GPR30 deficiency contributed to the rapid blood flow recovery in the arteries and capillaries, we did not elucidate the relative contribution of bicarbonate-induced GPR30 activation in SMCs and pericytes to ischaemia-reperfusion injury in vivo. Brain pericytes regulate microcirculation by organised contraction[38,39], and pericyte contraction leads to capillary constriction in the ischaemic penumbra in a mouse model of stroke[34,54]. The activation of $G_{\alpha q}$-coupled GPCRs triggers myofilament contraction in pericytes[55]. Pericyte contraction downstream of GPR30 activation by bicarbonate ions could be a mechanism underlying ischaemia-reperfusion injury. Pericyte and/or SMC contraction, subsequent blood flow reduction in wild-type mice, and the absence of these phenomena in GPR30-deficient mice are worth investigating in future research. Pericytes can be categorised into at least two forms by morphological and functional characteristics: contractile/ensheathing pericytes (*Acta2*+*Pdgfrb*+) and thin-strand pericytes (*Acta2*-*Pdgfrb*+). In the present study, we did not distinguish these two forms of pericytes, and the role of bicarbonate-GPR30 signalling in each form will be investigated in future studies. The mechanisms connecting the rapid blood flow recovery to improved outcomes in GPR30-deficient mice also remain unclear. Because GPR30 is expressed in different cell types in different organs, mural cell-specific GPR30-deficient mice would be more suitable for analysing the cerebral ischaemia-reperfusion model.

Pericyte-dependent capillary constriction is responsible for the 'no-reflow' phenomenon that occurs in clinical situations after recanalisation of occluded vessels following ischaemic stroke and myocardial infarction[56]. GPR30 signal will be one of the mechanisms involved in the 'no-reflow' phenomenon and a potential therapeutic target to avoid the constriction of capillaries in ischaemic stroke. GPCRs are the most intensively studied drug targets owing to their involvement in human pathophysiology and pharmacological tractability[57]. Modulation of GPR30 signalling may be an innovative therapy for cerebrovascular diseases.

Finally, we emphasise that we identified the first bicarbonate-sensing GPCR and suggest that pH resilience due to the bicarbonate buffer system modulates signal transduction via GPCRs. The bicarbonate buffer system not only maintains physiological pH but also works as a sophisticated mechanism that supplies bicarbonate ions and

protons to acid/base sensing GPCRs and modulates signal transduction in harmony with ever-changing extracellular environment.

## Methods
Our research complied with all relevant ethical regulations and the study protocols were approved by Safety Committee for Recombinant DNA Experiments, Radiation Safety Committee, and the Institutional Animal Care and Use Committee at Juntendo University School of Medicine.

### Database searches
Two-step selection of candidate GPCRs was implemented. The initial search used online data (https://pdsp.unc.edu/databases/ShaunCell/documents/RegardSupplementalGPCRExpressionRawData.xls)[14] linked with the PDSP database. Relative expression of 353 murine GPCRs in 41 tissues was analysed. Of the 353 murine GPCRs, 50 and 33 were highly expressed in the stomach and pancreas, respectively. From these, 10 GPCRs that were commonly highly expressed in both tissues were selected as potential candidates. The heatmaps with dendograms (Supplementary Fig. 1a, b) were constructed using online data 'Regard-Sato-Coughlin_Cell_Figure_3'[14]. The second search used public single-cell RNA-sequencing data of mouse brain vascular and perivascular cells (http://betsholtzlab.org/VascularSingleCells/database.html)[15,16].

### Cell line sources
The cell lines used in this study were COS-7 (ATCC, CRL1651); HEK293A (female origin; Thermo Fisher Scientific); MCF-7 (JCRB cell bank, JCRB0134); C2C12 (RIKEN BRC, RCB0987); and HepG2 (cDNA, a gift from Dr. Tanaka, The University of Tokyo). The cell lines are not in the mis-identification list.

### Vector construction and transfection
Human *Gpr30* cDNA was obtained from human hepatoblastoma-derived HepG2 cells. Mouse *Gpr30* was cloned using PCR from a mouse cDNA clone (#BC138598-seq, MGC-Premier, TCMS1004, TOT), and rat and zebrafish *Gpr30* was cloned using PCR from genomic DNA. The coding sequences of human and zebrafish *Gpr30* were inserted into the multi-cloning site of the plasmid vector pCXN2, which was generated in our laboratory via modification of pCAGGS[58], between the KpnI and EcoRI sites; the coding sequences of mouse and rat *Gpr30* were inserted into the multi-cloning site of pCXN2 between the NotI and EcoRI sites. The C-terminal HA-tagged *Gpr30* was amplified using a reverse primer containing the HA sequence.

One amino acid mutation of human GPR30 was generated as follows: the targeted amino acid was changed to alanine (GCC) using a two-step PCR method with the QuikChange® Primer Design Program by Agilent (https://www.agilent.com/store/primerDesignProgram.jsp), and the coding sequence with each mutation was inserted into the multi-cloning site of pCXN2 between the KpnI and EcoRI sites. The mutations generated were E115A, H120A, Q138A, H200A, E218A, E275A, H282A, H300A, H307A, and N310A. To obtain tagged GPR30 for detection of the receptor at the protein level, the mutated *Gpr30* was cloned using PCR with a reverse primer that added an HA tag to the C-terminus of GPR30. The ER-EGFP expression vector[59] was a gift from the Takayanagi laboratory at Kyushu University. The mock-EGFP vector was pEGFP-N2 from Clontech.

These vectors were transfected using the lipofection method (Lipofectamine™ 2000 Transfection Reagent, 11668019, Invitrogen; Lipofectamine™ 3000 Transfection Reagent, L3000015, Invitrogen). All cell lines and primary cultures were maintained at 37 °C and 5% CO$_2$.

### Live imaging
COS-7 cells expressing ERα-EGFP or mock-EGFP were treated with vehicle or 1 μM E$_2$ and maintained at 37 °C and 5% CO$_2$. Time-lapse imaging was performed using confocal microscopy (TCS SP5, Leica).

### TGFα shedding assay
The components and pH of all the buffers used in the study are shown in Supplementary Table 1. The TGFα shedding assay was performed according to a previously published protocol[26]. HEK293A cells (female origin; Thermo Fisher Scientific), their derivative G protein-deficient HEK293 (ΔG$_{q/11/12/13}$ −HEK293) cells, AP-TGFα expression vector, and chimeric G$_α$−subunit expression vectors were provided by Dr. Inoue and Dr. Aoki, Tohoku University.

HEK293 cells were seeded in 12-well plates at a density of $1 \times 10^5$ cells/well and cultured for 24 h. At 90% confluency, a mixture of plasmid vectors containing GPCR (see '***Vector construction and transfection***' section) and AP-TGFα, with or without G$_α$−subunit expression vectors, was transfected into the cells using Lipofectamine 2000 transfection reagent. After another 24 h of incubation, the cells were detached with 0.05% trypsin/EDTA (32777-44, Nacalai Tesque), suspended in Hanks' balanced salt solution (HBSS, Supplementary Table 1), and seeded in 96-well plates. The cells were stimulated for 1 h with $1.1$–$22 \times 10^{-3}$ M (final concentration) of NaHCO$_3$, $1 \times 10^{-14}$–$1 \times 10^{-6}$ M of E$_2$ (16156, Cayman Chemical; E1127, Sigma-Aldrich), or $1 \times 10^{-9}$–$1 \times 10^{-5}$ M of G-1 (10008933, Cayman Chemical), at 37 °C, under either 0.03% or 5% CO$_2$. Phorbol-12-myristate-13-acetate (PMA, 100 nM; #4174, Cell Signalling Technology) was used as a positive control. Conditioned media (CM) was transferred to another plate, and 80 μl of alkaline phosphatase (AP) solution (40 mM Tris-HCl, pH 9.5, 40 mM NaCl, 10 mM MgCl$_2$) was added to both plates, which were then incubated at 37 °C. The optical density at 405 nm (OD405) was measured using a microplate reader (iMark, Bio-Rad) at 5 min, 30 min, 1 h, and 2 h, depending on the reaction rate. The percentage of shed AP-TGF was calculated using the following equations:

$$AP\ activity = \triangle OD405\,(1-0\,h)$$

$$\%\ CM\ (conditioned\ media) = AP_{CM} / (AP_{CM} + AP_{Cell})$$

### Calcium assay
HEK293 or MCF-7 (JCRB cell bank, JCRB0134) cells ($2 \times 10^4$ cells/well) were seeded into a black wall and clear bottom 96-well plate 24–48 h before the assay. Cells at 90–100% confluency were incubated with HEPES buffer (1× HBSS, 2.5 mM probenecid, 20 mM HEPES, pH 7.4) containing 10 μM Fluo-8 AM (21080, AAT-Bio) for 60 min at 37 °C and 5% CO$_2$. The cells were washed twice with HEPES buffer. The cells were stimulated with indicated concentrations of various ligands including NaHCO$_3$, E$_2$, aldosterone (sc-210774, Santa Cruz), and G-1 at 37 °C and 0.03% CO$_2$. The fluorescence intensity was analysed using FlexStation 2 (Molecular Devices, Ex/Em = 490/525 nm). Where appropriate, cells were treated with PTX (100 ng/ml; List Biological Laboratories) for 16 h and with YM-254890 (1 μM; a gift from Dr. J. Takasaki[60]) for 45 min before the assay.

### Cyclic AMP assay
Cyclic AMP levels were measured using a CatchPoint™ cyclic AMP Fluorescent Assay Kit (R8088, Molecular Devices) via a competitive immunoassay for cAMP according to the manufacturer's instructions. Briefly, cells were seeded into a 96-well culture plate 24–48 h before the assay. The cells were stimulated with forskolin and bicarbonate for 15 min at 37 °C and then incubated with the lysis buffer for 10 min. The cAMP in the sample or standard competed with the horseradish peroxidase (HRP)-labelled cAMP conjugate for binding sites on anti-cAMP antibodies.

### Inositol phosphate accumulation assay
MCF-7 cells stably expressing mGPR30 or HEK293 cells transiently expressing hGPR30 were incubated with 1 μCi/ml myo-[³H] inositol

(NET114A, PerkinElmer) in growth medium for 24 h at 37 °C and 5% $CO_2$. Subsequently, cells were washed and incubated with 20 mM LiCl in HBSS for 20 min at 37 °C. Cells were then treated with vehicle or ligands for 30 min, and inositol phosphates were extracted using 5% perchloric acid. After neutralisation with 0.72 M KOH/0.6 M KHCO_3, centrifugation at 20,400 × $g$ for 10 min, and separation on Micro Bio-Spin™ Chromatography Columns (7326204, Bio-Rad) packed with AG 1-X8 Anion Exchange Resin (1432446, Bio-Rad), total inositol phosphates were eluted with a solution containing 0.1 M formic acid and 1 M formate. The radioactivity of the myo-[³H] inositol-containing phospholipids was measured using a liquid scintillation counter (Tri-Carb 5110TR, PerkinElmer).

### Selection of potentially essential key amino acids of GPR30 for bicarbonate recognition and/or downstream signalling of GPR30

Candidate amino acids required to recognise bicarbonate ions were selected based on the public homology model (https://gpcrdb.org/)[29]. Because there were many negatively charged amino acid residues deep inside the putative orthosteric pocket of GPR30, in which bicarbonate ions were unlikely to interact with amino acids, the conserved amino acids, from humans to zebrafish, among positively charged residues on the extracellular edge of its putative orthosteric pocket were first surveyed. The conserved amino acids included H200, H300, and H307. H307 was selected as a potential amino acid residue for recognition of bicarbonate ions because only the alanine substitution of H307 completely abolished the bicarbonate-induced activation of hGPR30. The candidates in the second step were hydrophilic amino acids located in the orthosteric pocket that could potentially cooperate with H307 to interact with bicarbonate ions.

### Analysis of cell surface expression of GPR30

The cell surface expression of wild-type and mutant hGPR30-HA was analysed via cell surface protein isolation using a Cell Surface Protein Isolation Kit (#89881, Thermo Scientific™) followed by western blotting. Briefly, HEK293 cells transiently expressing wild-type or mutant hGPR30-HA were biotinylated for 30 min at 4 °C. Then, the cells were harvested, and biotinylated proteins were isolated with avidin binding.

### Subcellular fractionation

HEK293 cells were harvested in sonication buffer (20 mM Tris-HCl, pH 7.4, 0.25 M sucrose, 10 mM $MgCl_2$, 2 mM EDTA-2Na, and protease inhibitor cocktail (25955-11, Nacalai Tesque)). Cells were centrifuged at 200 × $g$ for 10 min at 4 °C, and the supernatant was subjected to 3 min of sonication on ice. The lysate was centrifuged at 10,000 × $g$ for 10 min at 4 °C, and the pellet contained the subcellular membrane. The supernatant was then centrifuged at 100,000 × $g$ for 70 min at 4 °C, and the pellet contained the plasma membrane (PM). The supernatant was further centrifuged at 604,000 × $g$ for 90 min at 4 °C, and the pellet contained multivesicular bodies, including endosomes.

### Scintillation proximity assay

The PM fraction of HEK293 cells transiently expressing mock, hGPR30-HA, H307A-HA, E115A-HA, or Q138A-HA was used for the scintillation proximity assay (SPA). Protocol 1 (simultaneous addition protocol): the buffer (final concentration: 123 mM Tris-HCl, 2.5 mM $MgCl_2$, 0.62 mM EDTA), 11 mM sodium bicarbonate-[¹⁴C] (NEC086H, PerkinElmer), and 30 μg of the PM fraction were incubated for 15 min at RT. The SPA beads (WGA PVT SPA Scintillation Beads, RPNQ0252, PerkinElmer) were added and further incubated for 10 min with gentle agitation. Protocol 2 (pre-coupling method): 40 μg of the PM fraction and 2 mg of SPA beads were incubated for 4 h at 4 °C. Pre-coupled beads and the PM fraction were washed using the binding buffer (final concentration: 50 mM HEPES, 5 mM $MgCl_2$, 1 mM $CaCl_2$, 100 mM NaCl, and 5% bovine serum albumin [BSA], pH 7.5) and incubated with 11 mM sodium

bicarbonate-[¹⁴C] for 60 min with gentle agitation. Membrane-bound bicarbonate-[¹⁴C] was measured using a liquid scintillation counter (MicroBeta TriLux, PKI 1450, PerkinElmer). Data are presented as corrected count per minute (CCPM), where the count per minute (CPM) was normalised to the average CPM in the mock control group within the same experimental protocol.

### shRNA vector and knockdown cell line

The pSUPER RNAi System (VEC-PRT-0002, pSUPER.retro.puro, OligoEngine) was used for construction of the shRNA vector. The shRNA (Supplementary Table 2) was designed using an online RNAi design tool (http://www.oligoengine.com). Eight shRNA retroviral vectors were constructed according to the manufacturer's instructions. Among these candidates, one shRNA retrovirus vector was selected based on their knockdown efficiency after transient co-transfection with the target. Retroviruses carrying these shRNA vectors were generated in the packaging cell line Phoenix. Forty-eight hours after infection, cells were selected using puromycin (ant-pr-1, InvivoGen).

### Calcium imaging

C2C12 cells (3–6 × 10^5 cells/dish; RCB0987, RIKEN BRC) were seeded into a 35 mm glass-base dish (IWAKI) coated with collagen (Cellmatrix Type I-C, KP-4020, Kurabo) 72 h before the assay. Then, the cells were incubated in HBSS containing 2.5 μM Fura-2 AM (F-1201, Life Technologies) at 37 °C. The recording dish was washed twice with HBSS to stabilise the fluorescence ratio (340 nm/380 nm). Using a peristaltic pump at a flow rate of 1.5 ml/min, the ligand solution was applied sequentially to the cells for 15 s each at 4 min intervals.

Primary pericytes were seeded in collagen-coated 35 mm glass-bottom dishes (P35GCol-1.5-10-C, MatTek) and cultured for 3–7 days. The cells were incubated with 10 μM Fura-2 AM in growth media for 30 min at 37 °C and 5% $CO_2$. The recording dish was washed twice with HBSS to stabilise the fluorescence ratio (340 nm/380 nm). The ligand solution was applied sequentially to the cells for 3 min each at intervals of 7 min.

Freshly isolated SMCs and pericytes from the conditional GCaMP knock-in mice were seeded in collagen-coated, 35-mm glass-bottom dishes (P35GCol-1.5-10-C, MatTek) and incubated for 10 min at RT. The dish was centrifuged at 400 × $g$ for 2 min for the cells to attach. The recording dish was washed twice with HBSS, and the ligand solution was applied sequentially to the cells for 3 min each at intervals of 5 min. Intracellular calcium levels were monitored using an AQUA COSMOS $Ca^{2+}$ imaging system (Hamamatsu Photonics).

### RNA isolation and quantitative PCR

Total RNA was extracted using the acid guanidinium thiocyanate-phenol-chloroform extraction method with the Trizol reagent (15596-018, Life Technologies) or NucleoSpin RNA XS (740902.10, Macherey-Nagel). RNA quality was determined using a spectrophotometer (NanoDrop, NA-1000) and denaturing RNA electrophoresis. Reverse transcription was performed using a QuantiTect Reverse Transcription Kit (205311, Qiagen). Quantitative PCR was performed using a Fast SYBR Green Master Mix (4385612, Applied Biosystems) and normalised by the delta-delta Ct method. β-actin was used as the housekeeping gene. PCR amplification was carried out with initial denaturation at 95 °C for 20 s, followed by 40 cycles of 95 °C for 3 s and 60 °C for 30 s. Primer pairs are shown in Supplementary Table 3.

### Antibodies

The primary and secondary antibodies used in this study, along the validation statements from the manufacturer's website, are listed in Supplementary Table 4.

### Western blotting

Cell lysates were separated using sodium dodecyl sulphate-polyacrylamide gel electrophoresis under reducing conditions and

transferred to polyvinylidene difluoride membranes (Immobilon P, IPVH00010, Millipore). Primary antibodies used were phospho-p44/42 MAPK (Erk1/2) (Thr202/Tyr204), 1:1000 dilution, #9101, Cell Signalling Technology; ERK 2 (C-14), 1:1000 dilution, sc-154, Santa Cruz Biotechnology; anti-HA High Affinity, 1:1000 dilution, 11867423001, Roche; Na-K-ATPase, 1:1000 dilution, #3010, Cell Signalling Technology; Neuron-specific beta III tubulin (Clone Tuj-1), 1:1000 dilution, MAB1195, R&D systems; GAD67 [K-87], 1:1000 dilution, ab26116, abcam; Synaptophysin [YE269], 1:16000 dilution, ab32127, abcam; SPD-95 [EPR23124-118], 1:2000 dilution, ab238135, abcam; GFAP, 1:5000 dilution, #16825-1-AP, Proteintech; β-Actin (AC-15), 1:1000 dilution, sc-69879, Santa Cruz Biotechnology. Secondary antibodies used were anti-rabbit IgG, HRP-linked Antibody, 1:5000 dilution, #7074, Cell Signalling Technology; anti-mouse IgG, HRP-linked Antibody, 1:5000 dilution, #7076, Cell Signalling Technology; anti-rat IgG, HRP-Linked Whole Ab Goat, 1:5000 dilution, NA935, Cytiva. The membranes were probed at 4 °C overnight with the primary antibodies. The membranes were subsequently incubated with the corresponding secondary antibodies. The signals were detected with ECL Prime (RPN2236, Cytiva) or ImmunoStar LD (296-69901, FUJIFILM Wako Pure Chemical Corporation) using a chemiluminescence imaging system (ImageQuant LAS 4000mini, Cytiva; Fusion FX7, Vilber).

## Animals

Animal experiments were conducted in accordance with the ARRIVE guidelines[61]. All experimental protocols were approved by the Institutional Animal Care and Use Committee at Juntendo University (registration number: 1229, approved number: 270213, 280170, 290173, 300154, 310085, 2020148, 2021201, 2022201, and 2023210).

Mice were housed with 12 h–12 h light–dark cycles, an ambient temperature of 23 ± 2 °C, a humidity of 50 ± 10%, and an air change rate of 15/h. Mice were fed a standard chow diet (CRF-1, 15KGy, No.2108200, Oriental Yeast India Pvt. Ltd.) *ad libitum* and had free access to water. Cage bedding and chow diet were replaced weekly, and water was replaced twice a week. Microbiological monitoring was routinely performed once every three months. Euthanasia was executed by cervical dislocation, anaesthetic overdose, or $CO_2$ asphyxiation. All experiments were performed during the light phase of the cycle. The species, strain including substrain, sex, number, and age of animals used in every experiment, are listed in Supplementary Table 6.

## Generation of genetically modified mice

*Gpr30* knockout mice were generated using the CRISPR/Cas9 system on a C57BL/6 J (The Jackson Laboratory) background. *Gpr30-Venus* knock-in (*Gpr30-Venus*-KI) mice were generated on a C57BL/6 J (The Jackson Laboratory) background at Tsukuba University using CRISPR/Cas9 system[62]. The *Venus* sequence was inserted into the coding region of *Gpr30*, resulting in the expression of *Venus* under the *Gpr30* promoter. The expression of Venus was validated by analysing 10 μm frozen sections using confocal microscopy (TCS SP5, Leica). *Gpr30*[−/Venus] mice were generated by intercrossing *Gpr30* knockout mice and *Gpr30-Venus*-KI mice.

*Gpr30-iCre* knock-in (*Gpr30-iCre*-KI) mice were generated on a C57BL/6 J (C57BL/6JJmsSlc, Japan SLC, Inc.) background at National Institute for Physiological Sciences using CRISPR/Cas9 system. The *iCre* sequence was inserted into the coding region of *Gpr30*, resulting in the expression of *iCre* under the *Gpr30* promoter. The expression of *iCre* was validated using in situ hybridisation (Supplementary Fig. 6b–d). The conditional GCaMP knock-in mice were generated by intercrossing *Gpr30-iCre*-KI mice and B6-Gt(Rosa)26Sor<tm1(CAG-GCaMP6,-mCherry)Shi>[58,63] (RBRC09450, RIKEN) (Supplementary Fig. 6a).

## In situ hybridisation assay

The RNA in situ hybridisation assay was performed using the RNAscope technique (RNAscope® 2.5 HD Reagent Kit- RED, 322350, ACD;

RNAscope®Fluorescent Multiplex Reagent kit version 2, 323100, ACD), according to the manufacturer's instructions. Briefly, after deparaffinisation of 5 μm thick formalin-fixed paraffin-embedded sections, endogenous peroxidase blocking, antigen retrieval, and protease treatment, sections were hybridised with the target probes (RNAscope Probe-Mm-Gpr30, #318191; RNAscope Probe- Mm-Acta2-C2, RNAscope Probe-Mm-Pecam1-C2, #316721-C2; RNAscope Probe -Mm-Pdgfrb-C3, #411381-C3) for 2 h at 42 °C. Signal amplification, development, and counterstaining were performed according to the manufacturer's instructions.

## Quantification of pericyte and endothelial coverage

Heterozygous *Gpr30-Venus*-KI (*Gpr30*[+/Venus], *Gpr30*[−/Venus]) mice were perfused and fixed overnight with 4% paraformaldehyde. Fixed brains were sliced to 50 μm using a vibratome (VT1200S, Leica). Blood vessels were visualised by staining of collagen IV as a marker protein of the capillary basement membrane, pericytes by CD13 staining, and endothelial cells by CD31 staining[64]. The following antibodies were used: anti-CD13-Alexa 647 (5 μg/ml; #564352, BD Biosciences), anti-CD31-Alexa 647 (5 μg/ml; #102515, BioLegend), and rabbit anti-mouse collagen IV (1:500; #2150-1470, Bio-Rad) followed by Alexa 546-labelled donkey anti-rabbit IgG (4 μg/ml; A10040, Invitrogen). Brain sections were incubated in a blocking/permeabilisation solution (1% BSA [A2153, Sigma-Aldrich], 0.5% TritonX-100 in phosphate-buffered saline [PBS]) for 24 h at 4 °C, followed by incubation in primary antibody solution and subsequently in secondary antibody solution, both for 24–36 h at 4 °C. Sections were mounted in mounting medium (Vibrance Antifade Mounting Medium with DAPI, H-1800, Vector). Subsequently, 15–20 μm thick z-stacks were captured using confocal microscopy (LSM 880, Carl Zeiss AG). Five to six corresponding areas (right and left septal area, and frontal, parietal, and temporal cortex) in three or four animals of each genotype were analysed. Quantification of pericyte and endothelial coverage was performed using the ImageJ/Fiji software (NIH). The immunohistochemistry images presented are three-dimensional reconstructions of z-stacks using maximum intensity projection.

## Electron microscopy analysis

Mouse brains were fixed by perfusion with a fixed solution (2% PFA and 2% glutaraldehyde in phosphate buffer). The brain was sliced into 1 mm thick slices and subjected to additional fixation with 2.5% glutaraldehyde in 0.1 M phosphate buffer (pH 7.4). After post-fixation with 2% osmium tetroxide in the same buffer, these brain sections were dehydrated with a graded series of ethanol and embedded in Epok812 (Oken shoji). Ultrathin sections were cut with an ultramicrotome UC7 (Leica) and stained with uranyl acetate and lead citrate. These sections were analysed using a transmission electron microscope (HT7700, Hitachi) at an acceleration voltage of 100 kV.

## Isolation and primary culture of mouse brain vascular cells

The whole brain cortex was minced with scalpels and digested with collagenase/dispase (3.3 mg/ml; 11097113001, Roche) for 30 min at 37 °C. Myelin and debris were removed using Percoll spin (final 18%; P1644-500ML, Sigma-Aldrich) at 560 × g for 10 min at 4 °C. Brain pericytes were isolated as CD45−CD41−CD31−CD13+ cells using FACS (FACSAria IIIu, BD Biosciences)[30]. GPR30-positive brain vascular cells were harvested from heterozygous *Gpr30-Venus*-KI mice (Gpr30[+/Venus]) using the same procedure and isolated as Venus-positive cells using FACS (MoFlo Astrios, Beckman-Coulter).

## Fresh isolation of mouse brain vascular smooth muscle cells and pericytes

Individual SMCs were obtained from cerebral arteries, including the circle of Willis, the anterior cerebral artery, MCA, and posterior cerebral artery of the conditional GCaMP knock-in mice using an enzymatic

and mechanical dissociation procedure[65]. The isolated arteries were stored in isolation solution composed of 55 mM NaCl, 80 mM Na-glutamate, 5.6 mM KCl, 2 mM MgCl₂, 4 mM glucose, and 10 mM HEPES (pH 7.3). Papain (0.3 mg/ml) and DTT (0.3 mg/ml) were added, and the brain suspension was incubated for 14 min at 37 °C. The brain suspension was further incubated with in the isolation solution containing 0.67 mg/ml Collagenase F, 0.33 mg/ml Collagenase H, and 100 µM CaCl₂ for 5 min at 37 °C. Single cells were released by triturating 50 times with a fire-polished glass Pasteur pipette.

Capillary pericytes were isolated from the conditional GCaMP knock-in mouse brains using a papain-based Neural Tissue Dissociation kit (Miltenyi Biotec) and a mechanical dissociation procedure[65]. The whole brain cortex was chopped into small pieces with disposable scalpel; transferred to the enzyme mix 1 (Buffer X 2850 µl and enzyme P 75 µl/ a full adult brain) provided in the kit; and incubated for 17 min at 37 °C. Enzyme 2 was added, and the brain suspension was mixed 10 times using a Pasteur pipette and incubated for 12 min at 37 °C. The suspension was then passed 12 times through a 20-G needle and incubated for an additional 10 min. Myelin and debris were removed using Percoll spin (final 22%; P1644-500ML, Sigma-Aldrich) at 560 × g for 10 min at 4 °C. The suspension of isolated SMCs and pericytes was filtered through a 62-µm nylon mesh and stored in ice-cold isolation solution. The cells were used within 5 h after dispersion.

### Vascular leakage assay
Mice were injected with 3 kDa (10 µg/g body weight, injected volume 100–150 µl; D3308, Invitrogen), 10 kDa (15 µg/g, injected volume 100–150 µl; D1817, Invitrogen), or 70 kDa (100 µg/g, injected volume 150–200 µl; D1818, Invitrogen) dextran–Tetramethylrhodamine (TMR) dissolved in saline in the tail vein. After 3 (for 3 kDa dextran–TMR), 1 (for 10 kDa dextran–TMR), or 16 h (for 70 kDa dextran–TMR), the mice were anaesthetised with ketamine (0.1 mg/g body weight) and xylazine (0.01 mg/g body weight), perfused for 7 min with ice-cold PBS, and the brains were removed. After dissection, the brains were weighed and homogenised with 1% Triton X-100 in PBS. Brain lysates were centrifuged at 12,000 × g for 20 min at 4 °C, and the supernatant was used to quantify fluorescence (Ex/Em 540/590 nm; FlexStation 3, Molecular Devices). The relative fluorescence values were normalised to the brain weights.

### MCAO modelling
We subjected only male mice to MCAO to evaluate the effects of GPR30 in ischaemia-reperfusion injury because previous studies suggested that premenopausal females are protected against capillary dysfunction and brain injury in particular conditions[35]. Mice were anaesthetised with 5.0% isoflurane (Abbott Japan Co., Ltd., Tokyo, Japan) and maintained with 2.0% isoflurane in 70% N₂O and 30% O₂ using a small-animal anaesthesia system. Transient cerebral focal ischaemia was induced using the intraluminal filament technique[66]. A silicon-coated nylon monofilament (handmade filament with coating tip diameter 0.18 mm and length 8–9 mm for Fig. 7 and Supplementary Fig. 8; Doccol, 602045PK10, or 602256Re for Fig. 8 and Supplementary Fig. 9) was inserted through the left internal carotid artery immediately after ligation of the ipsilateral common and external carotid arteries. After 60 min of occlusion of the left MCA, recanalisation was accomplished by pulling the filament out of the artery. Neurological severity was assessed on a scale of 0 (normal) to 18 (maximal deficit) using the mNSS, which is a composite of motor (muscle status, abnormal movement, and balance), sensory (visual, tactile, and proprioceptive), and reflex test scores[67].

### IgG staining
At 3 or 7 days after MCAO, the brains were perfused and fixed overnight with 4% paraformaldehyde. Fixed brains were sliced into 20 µm

thick sections using a cryostat (CM3050S, Leica). The brain sections were treated with blocking solution (10% horse serum in PBS) for 1 h at 20–25 °C, incubated overnight at 4 °C with the biotinylated anti-mouse IgG (H + L) (1:100; BA-2000, Vector Laboratories) in 10% horse serum in PBST. The VECTASTAIN Elite ABC Standard Kit (PK-6100, Vector Laboratories) was used as a signal enhancer, and DAB colour development was performed. Images were taken, combined, and analysed using a digital microscope (BZ-X800 Viewer/Analyzer 1.1.2.4, KEYENCE). IgG staining was quantified by integrating the intensity values of the 12 sections from each mouse.

### Cresyl violet staining
At 3 or 7 days after MCAO, the brains were perfused and fixed overnight with 4% paraformaldehyde. Fixed brains were sliced into 20 µm thick sections using a cryostat (CM3050S, Leica). The brain sections were incubated in a cresyl violet staining solution (0.1% cresyl violet solution (41021, Muto Pure Chemicals) with 0.15% acetic acid) for 8 min at 37 °C, followed by differential staining with 100% EtOH. Images were taken, combined, and analysed using digital microscopy (BZ-X800 Viewer/Analyzer 1.1.2.4, KEYENCE). Cresyl violet staining was quantified by integration of the cresyl violet-negative area of 12 sections from each mouse.

### TUNEL assay
TUNEL staining was performed using 20 µm thick frozen brain sections according to the manufacturer's instructions (In Situ Cell Death Detection Kit, TMR red, #12156792910, Merck). Subsequently, 1–4 µm thick z-stacks were captured, reconstructed into fully focused images, and combined with whole brain images using digital microscopy (BZ-X800 Viewer/Analyzer 1.1.2.4, KEYENCE). Quantification of TUNEL-positive cells was performed on the overall left hemibrain using ImageJ software (NIH).

### Serum collection and measurement of ion concentration
Blood was collected from the facial vein before and after reperfusion of the MCAO. The samples were left at 20–25 °C for coagulation, centrifuged at 5000 × g for 20 min, and used for further analysis. The bicarbonate concentration was measured using a commercially available kit (KTCO-801, TOYOBO). Briefly, phosphoenolpyruvate carboxylase (PEPC) catalyses the reaction of bicarbonate with phosphoenolpyruvate (PEP), resulting in the production of oxaloacetic acid. Oxaloacetic acid is reduced to malic acid by malate dehydrogenase (MDH), while acetyl nicotinamide adenine dinucleotide (aNADH) is oxidised to aNAD⁺. In these reactions, the decline in NADH absorption of aNADH is parallel to the concentration of bicarbonate ions. Samples were added to the reaction mixture containing PEP, PEPC, aNADH, and MDH, incubated for 5 min at 37 °C, and analysed using a spectrometer (SpectraMax M2e, Molecular Devices). The concentrations of calcium, magnesium, phosphate, sodium, and chloride were measured using FUJI DRI-CHEM (Fujifilm).

### MRA analysis
All magnetic resonance imaging (MRI) experiments were performed using the 7 T Bruker preclinical MRI system (Biospec 70/16; Bruker BioSpin, Ettlingen, Germany) and cryogenic quadrature radiofrequency surface probe (CryoProbe; Bruker BioSpin AG, Fällanden, Switzerland). During the MRI experiments, mice were anaesthetised under 1.2–1.5% isoflurane with continuous monitoring of the respiratory rate. Body temperature was continuously maintained above 34.0 °C by circulating water through heating pads. MRA scans were acquired before and 30 min after reperfusion using a 3D time-of-flight sequence with the following scanning parameters: TR/TE 30/2.38 msec, average 1, flip angle 40, FOV 12.8 × 12.8 × 12.8 mm, and matrix 192 × 192 × 192. Branching vessels in the right and left MCA regions

after reperfusion were delineated based on the Canny edge detection method[68] implemented in MATLAB (R2017b, Mathworks, MA, USA) and then manually segmented. The cerebrovascular volume of the left hemisphere (reperfusion side) was evaluated as a ratio to that of the right hemisphere (control side).

### Laser doppler flowmetry

Laser Doppler flowmetry (OMEGAFLO, FLO-C1, OMEGAEAVE) was used to continuously monitor cerebral blood flow changes before and during MCA occlusion and 20 min after reperfusion. The laser Doppler probe was glued to the surface of the left side of the skull at a point 1.0 mm posterior and 5.0 mm lateral to the bregma before the operation. The cerebral blood flow was continuously monitored before the MCAO surgery (basal blood flow), during common carotid artery (CCA) occlusion, MCAO, and reperfusion. Cerebral blood flow was normalised as a ratio to the basal blood flow.

### Statistics and reproducibility

All experiments were conducted with technical and biological replicates (Supplementary Tables 5, 6). No statistical methods were used to predetermine the sample size. No methods of randomisation were applied but blinding was applied during the MCAO procedure, mNSS scoring (Fig. 7b, Supplementary Fig. 8a), and MRA analysis (Fig. 8d, e). Dot plots were routinely used to show individual data points. Statistical analyses were performed using GraphPad Prism 9 (GraphPad Software). Two-sided tests were used in all statistics. $P$ values < 0.05 were considered significant; exact $P$ values are provided. $P$ values > 0.05 were considered non-significant and exact values are shown where appropriate. The statistical tests used are stated in each figure legend.

### Reporting summary

Further information on research design is available in the Nature Portfolio Reporting Summary linked to this article.

## Data availability

Source data are provided with this paper. The data generated in this study are provided in the Supplementary Information/Source Data file. The raw data generated in this study have been deposited in Figshare under https://doi.org/10.6084/m9.figshare.24431842[69]. Accession code of mouse Gpr30 mRNA is NM_029771. The web links for publicly available datasets analysed during the current study are as follows: murine GPCR expression data are available from https://pdsp.unc.edu/databases/ShaunCell/documents/RegardSupplementalGPCRExpressionRawData.xls (the Psychoactive Drug Screening Program (PDSP) database)[14]; murine GPR30 expression data are available from http://biogps.org/#goto=genereport&id=76854 (BioGPS); human GPR30 structure model is available from https://gpcrdb.org/protein/gper1_human/? (GPCRdb); and the single cell RNA-sequencing data of mouse brain vascular and perivascular cells are available from http://betsholtzlab.org/VascularSingleCells/database.html[15,16]. Source data are provided with this paper.

## Code availability

The custom computer code used in the current study is deposited in the Figshare under https://doi.org/10.6084/m9.figshare.24431842[69].

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

## Acknowledgements

We thank the members of the Yokomizo laboratory and other faculty members at Juntendo University for their comments on the results; A. Inoue and J. Aoki for providing the TGFα shedding assay system; S. Mizuno, S. Takahashi, and Laboratory Animal Resource Center in Transborder Medical Research Center, University of Tsukuba, for generating *Gpr30-Venus*-KI mice; M. Hirabayashi, M. Sanbo, and Section of Mammalian Transgenesis, Center for Genetic Analysis of Behaviour, National Institute for Physiological Sciences, for generating *Gpr30-iCre*-KI mice; J. Miyazaki for the GCaMP knock-in mice; Y. Kamikubo for help with brain primary culture; S. Kakuta for help with electron microscopy and confocal microscopy analyses; K. Yamashiro for analyses of the MCAO model; and the members of Laboratory of Morphology and Image Analysis, Laboratory of Molecular and Biochemical Research, Laboratory of Cell Biology, Laboratory of Biomedical Research Resources, Laboratory of Radioisotope Research, and Laboratories of Proteomics and Biomolecular Science, Biomedical Research Core Facilities, Juntendo University, for technical assistance. This work was supported by MEXT/JSPS KAKENHI (grant numbers JP18K15051 and 20K16148 to A.J.-W.; JP18H02627, JP19KK0199, and 21H04798 to T.Y.) and the Japan Agency for Medical Research and Development (Innovation and Clinical Research Center Project, PRIME, grant number JP20gm6210026 to A.J.-W.).

## Author contributions

A.J.-W. conceived the project and designed and performed all the experiments with inputs from T.Y. T.I. constructed and guided the MCAO model under the supervision of N.H. T. Osada performed the analyses of MRA data. R.H. and A.J.-W. conducted calcium imaging of the primary cultures. A.J.-W., S.I. and K.T. performed the knockdown assay and calcium imaging of the cell lines. T.N. and O.N. developed the homology models. T. Okuno provided advice on the in vitro experiments. M.O.-H. provided advice on the in vivo experiments. A.J.-W., T.Osada, M.O.-H. and T.Y. wrote the manuscript with inputs from all authors. All authors discussed the results, reviewed, and approved the final manuscript.

## Competing interests

The authors declare no competing interests.
