## [Peer Review File · Nature Communications]

REVIEWER COMMENTS

Reviewer #1 (Remarks to the Author):

This is an interesting manuscript that reports the observation that GPR30, heretofore considered an estrogen sensing GPCR, may in fact be a Gαq-linked receptor for HCO₃. In addition, GPR30 expression in the brain is identified with pericytes, which respond to HCO₃ with elevated intracellular Ca, and deletion of GPR30 is found to improve reperfusion blood flow and reduce neurological severity after ischemia-reperfusion in mice.

Overall, this paper provides a comprehensive and compelling demonstration that GPR30 can respond to changes in HCO₃, both in heterologous expression systems and in native cells; it also incorporates a characterization of molecular determinants required for HCO₃ activation of GPR30. A further strength of the paper is the development and use of multiple lines of GPR30 mutant mice, including one with a Venus-tagged GPR30 allele, for identifying cell types expressing GPR30 and to reveal a role for GPR30 in ischemia reperfusion injury. The principal shortcoming of the work, as outlined below in the major concern, is that there is no direct demonstration that HCO₃ serves as a relevant agonist for GPR30 in vivo and the concentration-response curves instead imply that there would be little effect at the levels of HCO₃ that are encountered under physiological conditions.

Major:

1. As mentioned, the major problem with the conclusions relates to the EC₅₀ for HCO₃ activation of GPR30 (~11-12 mM). According to the concentration-response curves, it appears that GPR30 signaling is essentially saturated at physiological concentrations of HCO₃ (i.e., ~22-25 mM) and thus it is not clear how the small increase above those concentrations elicited by reperfusion would be able to activate GPR30 any further (i.e., from ~22 mM to ~26-27 mM; Fig. 5b). In fact, given the position of the physiological serum HCO₃ concentration relative to the EC₅₀, it would appear that this receptor would be more likely to signal decreases in HCO₃ concentration (by decreased receptor activation) rather than increases of the sort measured here during ischemia-reperfusion.

This does not negate the data from mice showing that GPR30 can influence blood flow during reperfusion and the severity of IRI, but it does call into question whether the effect is due specifically to HCO₃ sensing by GPR30. It is possible that local changes in [HCO₃] could be in the appropriate range, but we are not provided with any evidence for such relevant local concentrations. This issue should be resolved and/or explicitly acknowledged and discussed as a limitation.

Minor:

1. Please specify the pH of all the buffers used to test functional effects of HCO₃ on GPR30-expressing cells.

2. What is the value of including the GPR30-/Venus mouse? Isn't it functionally equivalent to GPR30Venus/Venus?

3. Extended Fig. 4: H200 in panel b, but elsewhere data/text says H282.

4. The data with GPR30 point mutants would be more convincing if cell surface expression was determined for all the variants (e.g., by cell surface biotinylation and avidin IP).

5. l. 187: "these three amino acids are essential for bicarbonate recognition and downstream signalling of GPR30" ... should probably be "and/or" since it is not clear whether those residues contribute to either binding or receptor signalling or both.

6. What is HA-hBLT1 (used in Fig. 1i) – is this a leukotriene B receptor used as a negative control for scintillation proximity assay (SPA)? It might have been informative to attempt this assay with the His-mutated GPR30 to see if HCO₃ binding was affected (see point #5).

7. l. 238: "GPR30 drives ischaemia-reperfusion injury": would suggest rewording since "drives" is an overstatement.

8. l. 324: "we focused on GPR30, which is exclusively expressed in the brain microvasculature ..." The use of "exclusively" is incorrect since it is also expressed in kidney, and it is generally advisable to be cautious with such statements.

9. The framing of the paper seems to be a bit confused. On the one hand, the discovery of HCO₃ as a receptor agonist is treated as serendipity, but on the other hand the first sections of the manuscript are presented as a concerted bioinformatics effort to find such a HCO₃-activated receptor. So, the reader may be left wondering how this came to pass: Was it simply good luck and careful observation? or good sleuthing based on an insightful hunch?

Reviewer #2 (Remarks to the Author):

In this manuscript, authors have identified the first bicarbonate-sensing GPCR and suggest that pH resilience due to the bicarbonate buffering system modulates signal transduction via GPCRs. The local concentration of bicarbonate ions rapidly increases upon reperfusion, which activates GPR30-positive pericytes, leading to the inhibition of the rapid recovery of peripheral circulation. The manuscript is well written, easy to read, and within the scope of the submitted journal. However, additional experiments and information should be done to improve the quality of the manuscript. Some concerns are outlined below, hoping that it helps the authors strengthen the manuscript.

1. Local pH and ion homeostasis are dynamically changed, and could be influenced by the microenvironment. How would the authors to control other variances?
2. Why “10 of 353 GPCRs examined were selected based on their predominant expression in the stomach and pancreas”, but the object of this study is the brain?
3. Please clarify the ER-EGFP and mock-EGFP?
4. The word size in Figures is too small to see. Please make all words bigger.
5. In Figure 3, have the authors checked the GPR30 expression on SMCs?
6. The GPR30-deficient mice were utilized in this study. Will the pericyte conditional GPR30-deficient mice be more specific to explore the functions of GPR30?
7. To show mature and functional neurons, was there synapse protein expression in the bioprinted neural tissue.

Reviewer #3 (Remarks to the Author):

The manuscript by Jo-Watanabe and colleagues describes the results of the experimental studies collectively suggesting that bicarbonate signalling via GPR30 expressed by pericytes contributes to ischaemia-reperfusion brain injury. The authors propose that ischemia is associated with an increase in extracellular bicarbonate, which acts at GPR30 expressed by pericytes, leading to Gq mediated elevation of intracellular calcium, pericyte constriction, ultimately preventing successful reperfusion of the affected brain region. The study appears to be expertly performed, nicely illustrated and the text of the manuscript is well written. The data reported may prove to be highly significant and suggest a novel druggable target for the treatment of ischaemic stroke. I only have two comments to make which the authors may consider in their revision:

1. Fig 1g shows that the responses (IP accumulation) mediated by GPR30 expressed in HEK293 cells are pretty much saturated at concentrations above 20 mM. Fig 2a shows similar responses (Ca²⁺ mobilization) to 22 and 33 mM HCO₃⁻ in mouse myoblasts. Fig 3i shows robust Ca²⁺ responses in brain pericytes triggered by 11 and 22 mM bicarbonate. If these observations are correct, then GPR30 expressed by pericytes would be constitutively active at physiological concentrations of extracellular HCO₃⁻ in the brain (22-26 mM). Fig 5b,c shows a very modest increase (by ~4 mM) in bicarbonate measured in the blood collected from the facial vein. The authors need to discuss how they envisage the operation of the proposed mechanism as the receptor is clearly strongly activated by bicarbonate at its physiological concentration and ischaemia/reperfusion-induced increases in HCO₃⁻ are too small to trigger functionally significant further increases in pericyte Ca²⁺.

2. I suggest that authors discuss the potential mechanisms leading to the increased concentration of extracellular bicarbonate during ischaemia/reperfusion.

Reviewer #4 (Remarks to the Author):

This is an excellent study which identifies the bicarbonate sensing role of GPR30 in pericytes as a key modulator of blood flow in ischemia reperfusion. The systematic testing which establishes the bicarbonate response of the receptor while ruling out other potential contributors is quite beautifully done. I have several comments that I hope the reviewers will address to strengthen the manuscript.

1. As the authors acknowledge, GPR30 is also highly expressed in smooth muscle cells, which is evident in the RNAseq database they utilize and likely also shown in their images of GPR30 localization. It is important to delineate responses of pericytes vs. SMCs to bicarbonate levels with higher resolution to determine relative contributions to bicarbonate fluctuations in vivo. This could be achieved by imaging calcium in these cells either isolated from the brain and loaded with calcium dyes, or in cells isolated from mice expressing GCaMP under the NG2 or pdgfrb promoters. The smooth muscle may play an important role here that is overlooked, as changes in bicarbonate during I-R injury are likely to cover large areas which encapsulate both pericytes on capillaries and SMCs on arterioles.

2. The use of primary culture to demonstrate the bicarbonate signal in brain pericytes is not ideal, as phenotypic drift may affect responses. Use of an acute isolation protocol e.g. PMID: 35349300 would increase confidence that activation of native GPR30 in pericytes by bicarbonate ions is a physiologically relevant event.

3. Minor comment: At times figure labeling is unclear, and a careful check that all terms are defined in the figure legends is important for readability/interpretability. For example, Ex Fig 7d-f have an 'NT' condition which is not defined which makes interpretation difficult.

4. Minor comment: A number of figures also have a confusing star denoting significant differences between groups in the legend, but then testing on the data show no difference. For example Ex Fig 7b has a star between group labels, but then both tests on the data show 'ns' for not significant. This issue is present in several figures and creates confusion. Clarification in the figure legends or some other approach is needed to avoid confusing the reader.

Responses to reviewers' comments on 'Bicarbonate signalling via GPCR regulates ischaemia-reperfusion injury' by Jo-Watanabe et al. (NCOMMS-23-03129-A).

Responses to Reviewer #1

We greatly appreciate Reviewer #1 for taking the time to evaluate our manuscript.

We provide point-by-point responses to the reviewer's comments. Unless otherwise specified, the figure, page, and line numbers refer to those in the revised manuscript. Modified text is shown in blue. Text in red indicates modifications for multiple reviewers.

This is an interesting manuscript that reports the observation that GPR30, heretofore considered an estrogen sensing GPCR, may in fact be a Gαq-linked receptor for HCO₃. In addition, GPR30 expression in the brain is identified with pericytes, which respond to HCO₃ with elevated intracellular Ca, and deletion of GPR30 is found to improve reperfusion blood flow and reduce neurological severity after ischemia-reperfusion in mice.

Overall, this paper provides a comprehensive and compelling demonstration that GPR30 can respond to changes in HCO₃, both in heterologous expression systems and in native cells; it also incorporates a characterization of molecular determinants required for HCO₃ activation of GPR30. A further strength of the paper is the development and use of multiple lines of GPR30 mutant mice, including one with a Venus-tagged GPR30 allele, for identifying cell types expressing GPR30 and to reveal a role for GPR30 in ischemia reperfusion injury. The principal shortcoming of the work, as outlined below in the major concern, is that there is no direct demonstration that HCO₃ serves as a relevant agonist for GPR30 in vivo and the concentration-response curves instead imply that there would be little effect at the levels of HCO₃ that are encountered under physiological conditions.

Major:

- 1. As mentioned, the major problem with the conclusions relates to the EC₅₀ for HCO₃ activation of GPR30 (~11–12 mM). According to the concentration-response curves, it appears that GPR30 signaling is essentially saturated at physiological concentrations of HCO₃ (i.e., ~22–25 mM) and thus it is not clear how the small increase above those concentrations elicited by reperfusion would be able to activate GPR30 any further (i.e., from ~22 mM to ~26–27 mM; Fig. 5b). In fact, given the position of the physiological serum HCO₃ concentration relative to the EC₅₀, it would appear that this receptor would be more likely to signal decreases in HCO₃ concentration (by decreased receptor activation) rather than increases of the sort measured here during ischemia-reperfusion.*

This does not negate the data from mice showing that GPR30 can influence blood flow during reperfusion and the severity of IRI, but it does call into question whether the effect is due specifically to HCO₃ sensing by GPR30. It is possible that local changes in [HCO₃] could be in the appropriate range, but we are not provided with any evidence for such relevant local

concentrations. This issue should be resolved and/or explicitly acknowledged and discussed as a limitation.

Response: We acknowledge the validity of your concern. We recognize that bicarbonate-induced activation of GPR30 *in vivo* requires shifting local bicarbonate levels over the dynamic range, as demonstrated in Fig. 2h and i of the revised manuscript. The change in the systemic, i.e., serum, bicarbonate levels does not precisely mirror the change in local bicarbonate levels in the brain, although the bicarbonate concentration in the serum sometimes reflects that in the brain. The systemic attenuation of the local dynamic shift in the bicarbonate concentration is attributed to the bicarbonate-buffering system *in vivo*.

Previous studies consistently demonstrated that ischaemia causes a significant change in the local acid–base balance in the brain: a decrease in pH (from 7.3 to 6.0–6.5), an increase in pCO₂ (from 40–50 to >100 mmHg), and a decrease in [HCO₃⁻] (from 20–26 to 11–12 mM extracellularly and 11–12 to 4–8 mM intracellularly). Importantly, reperfusion restores these deviations to the pre-ischemic state (¹Zha et al., 2022. doi: 10.1177/0271678X221089074; ²Smith et al., 1986. doi: 10.1038/jcbfm.1986.104; ³Kawabata, 1993. doi: 10.11482/KMJ19(1)25-35.1993.pdf).

Figure A1. Change in cerebrospinal fluid (CSF) pH, PCO₂, and HCO₃⁻.
Each point indicates the mean ± SE for six dogs.
#: Significantly different from pre-ischemic value. (p<0.05)

Figure A2. Change in arterial pH, PCO₂, and HCO₃⁻.
Each point indicates the mean ± SE for six dogs.
#: Significantly different from pre-ischemic value. (p<0.05)

Adapted from Kawabata Y, ‘Effect of Tris-hydroxymethyl-aminomethane on Arterial Blood, Brain and Cerebrospinal Fluid Acidosis after Total Cerebral Ischemia in Dogs’, *Kawasaki Medical Journal*, 1993.

Contrastingly, the systemic pH, pCO₂, and [HCO₃⁻] do not shift as much as those in the local microenvironment. Previous studies which parallelly evaluated CSF/tissue and blood bicarbonate concentrations demonstrated that the CSF/tissue bicarbonate concentrations decreased to 11–12 mM, equivalent to the EC₅₀ in our *in vitro* experiment, during ischaemia and recovered to the preischemic level by reperfusion, whereas ischaemia caused a slight decrease (~3 mM) in the blood bicarbonate

concentrations (²Smith et al., 1986. doi: 10.1038/jcbfm.1986.104; ³Kawabata, 1993. doi: 10.11482/KMJ19(1)25-35.1993.pdf, **Figure A1 and A2**). Therefore, in our experiment, it is most likely that **the extracellular bicarbonate concentration decreased to the EC₅₀ of bicarbonate-induced GPR30 activation (11–12 mM) during ischaemia** and recovered enough for full activation (20–26 mM) after reperfusion, while the serum bicarbonate concentration increased slightly but significantly from 22 to 26 mM. **Consequently, changes in the local bicarbonate concentration during ischaemia and reperfusion likely affected GPR30 activation *in vivo*.** It should be noted, however, that we cannot exclude the possibilities that the *in vivo* dynamic range of GPR30 activation by bicarbonate is different from that *in vitro* and that constitutive activation of GPR30 is involved in the pathophysiology of ischaemia-reperfusion injury.

The restricted blood flow limits the supply of oxygen to the tissues. Accordingly, the cellular metabolic state shifts from aerobic respiration to anaerobic glycolysis, producing lactate that causes intracellular acidosis. The following mechanisms to counteract intracellular acidosis causes interstitial acidosis. One is the subsequent lactate export via monocarboxylate transporters, resulting in extracellular metabolic acidosis. Other mechanisms include activation of the Na⁺/H⁺ exchanger, NHE, and import of bicarbonate ions to counteract intracellular acidification, via SLC4A family transporters: sodium bicarbonate cotransporter NBC, Na⁺/Cl⁻/2HCO₃⁻ cotransporter/exchanger NDCBE, and Cl⁻/HCO₃⁻ exchanger AE (⁴Choi, 2012. doi: 10.1016/B978-0-12-394316-3.00003-X). **All these mechanisms contribute to the reduction of extracellular bicarbonate levels** (⁵Chesler, 2003. doi: 10.1152/physrev.00010.2003).

The above-mentioned studies used rat or canine ischaemia-reperfusion models and calculated [HCO₃⁻] from the pH and pCO₂ values that were measured using electrodes inserted in the brain. Because direct measurement of extracellular bicarbonate concentrations in mice would be informative, we attempted to directly quantify local changes in the bicarbonate concentration using a microdialysis system in the mouse MCAO model. *In vitro* recovery tests demonstrated that the bicarbonate concentration of the recovered fluid mirrored the local concentration outside the dialysis membrane (**Figure B1**). However, the recovered fluid from the MCAO mice consistently contained the same concentration of bicarbonate as the dialysate over a range of bicarbonate concentrations (**Figure B2**), indicating that our microdialysis system failed to reflect the local bicarbonate concentration *in vivo*.

Figure B1. *In vitro* recovery test.

A microdialysis probe was sequentially inserted into the tubes containing different sodium bicarbonate concentrations. Bicarbonate concentrations of dialysate, sodium bicarbonate solutions in the tubes (outside), and recovered fluid were measured.

Figure B2. *In vivo* microdialysis.

A microdialysis probe was inserted into the left striatum. Dialysates containing different sodium bicarbonate concentrations were continuously infused, and recovered fluid was collected. Bicarbonate concentrations of dialysate and recovered fluid were measured.

In the revised manuscript, we explicitly acknowledged that we did not measure the local bicarbonate concentration as a limitation of this study in the Discussion section. We also discussed whether the changes in [HCO₃⁻] in the brain could be in the appropriate range to cause GPR30 activation and how the local acid–base balance and electrolytes change during ischaemia and reperfusion (lines 389–420).

Minor:

1. Please specify the pH of all the buffers used to test functional effects of HCO₃ on GPR30-expressing cells.

Response: We added all the pH values of the used buffers in the Supplementary Table 3.

2. What is the value of including the GPR30-Venus mouse? Isn't it functionally equivalent to GPR30Venus/Venus?

Response: *Gpr30*^{Venus} mice are functionally equivalent to *Gpr30*^{Venus/Venus} and *Gpr30*^{-/-} mice. To evaluate pericyte and endothelial coverage, we first used FITC-labelled anti-CD13 antibody. Because the absorption and emission wavelengths of Venus overlap those of FITC, it was difficult to accurately distinguish the expression of Venus and CD13. Therefore, we used heterozygous *Gpr30-Venus* knock-in mice (*Gpr30*^{+Venus} and *Gpr30*^{-Venus}), which are expected to express similar levels of Venus. However, along with the modification of the immunohistochemistry protocol, in this manuscript, we used Alexa 647-labelled, instead of FITC-labelled, anti-CD13 antibody to achieve an accurate evaluation. We have mentioned this issue in the Results section (lines 261–264).

3. Extended Fig. 4: H200 in panel b, but elsewhere data/text says H282.

Response: We apologize for the incomplete description of the selection steps and have added the phrase '**could potentially cooperate with H307 in the interaction with bicarbonate ions**' in lines 179-180. We have also added how we selected the candidate amino acids in the Results (lines 169–178) and Methods (lines 553–566) sections.

Extended Fig. 4b in the previous manuscript (Fig. 3b in the revised manuscript) shows the primary candidates for recognising bicarbonate ions according to the public homology model (<https://gpcrdb.org/>). Many negatively charged amino acid residues were located deep inside the putative orthosteric pocket of GPR30, where bicarbonate ions are unlikely to interact with amino acids. Therefore, we first surveyed positively charged residues on the extracellular edge of its putative orthosteric pocket (Fig. 3a). The conserved amino acids from humans to zebrafish, as zebrafish GPR30 was activated by bicarbonate ions as shown in Supplementary Fig. 21, include H200, H300, and H307 (Fig. 3b). H307 was selected as a potential amino acid residue for recognising bicarbonate ions because the alanine substitution of H307 only completely abolished the bicarbonate-induced activation of hGPR30 (Fig. 3c, d).

Additionally, H282 is one of the candidates in the second step that are hydrophilic amino

acids located in the orthosteric pocket and **could potentially cooperate with H307 in the interaction with bicarbonate ions** (Fig. 3e). H282 is not conserved in zebrafish, and the alanine substitution of H282 did not affect the bicarbonate-induced activation of hGPR30 (Fig. 3c, d).

4. *The data with GPR30 point mutants would be more convincing if cell surface expression was determined for all the variants (e.g., by cell surface biotinylation and avidin IP).*

Response: According to your comment, we evaluated the cell surface expression of all the mutants using cell surface biotinylation and avidin IP (Pierce & trade; Cell Surface Protein Isolation Kit, Thermo Scientific™, #89881). Western blotting of HA-tagged mutants indicated that the cell surface expression levels of mutated receptors with H307, E115, and Q138, essential for bicarbonate recognition and/or downstream signalling, were comparable to all other mutants. These results have been added in the Results section (lines 185–186) and Supplementary Fig. 3a, b. We have also added the procedure in the Methods section (lines 567–573).

5. *l. 187: “these three amino acids are essential for bicarbonate recognition and downstream signalling of GPR30” ... should probably be “and/or” since it is not clear whether those residues contribute to either binding or receptor signalling or both.*

Response: We agree with your comment that ‘and’ should be ‘and/or’ in the sentence, as it is not clear whether these residues contribute to either binding or receptor signalling or both. Therefore, we have corrected the sentence: ‘these three amino acids are essential for bicarbonate recognition **and/or** downstream signalling of GPR30’ in line 187 and added the sentence ‘H307 is likely to be involved in the recognition of bicarbonate ions’ in lines 197–199.

6. *What is HA-hBLT1 (used in Fig. 1i) – is this a leukotriene B receptor used as a negative control for scintillation proximity assay (SPA)? It might have been informative to attempt this assay with the His-mutated GPR30 to see if HCO₃ binding was affected (see point #5).*

Response: Per your suggestion, we performed the SPA assay with the three mutants involving amino acids essential for bicarbonate recognition and/or downstream signalling of GPR30. The results showed that wild-type GPR30 caused a higher SPA count than the mock. As expected, the H307A mutant caused a lower SPA count than wild-type GPR30 and the other mutants. The results of the SPA assay have been modified and shown in the Results section (lines 189–192, 194–196) and Supplementary Fig. 3c, d. Although the data were not conclusive, they are consistent with the results of the procedure of selecting candidates for recognition of bicarbonate ions (lines 197–199).

7. *l. 238: "GPR30 drives ischaemia-reperfusion injury": would suggest rewording since "drives" is an overstatement.*

Response: Per your suggestion, we have changed the word ‘drives’ to ‘contributes to’ in line 252.

8. *l. 324: "we focused on GPR30, which is exclusively expressed in the brain microvasculature ..." The use of "exclusively" is incorrect since it is also expressed in kidney, and it is generally advisable to be cautious with such statements.*

Response: We apologize for our inappropriate word usage. According to your suggestion, we have omitted the word ‘exclusively’ in line 346.

9. *The framing of the paper seems to be a bit confused. On the one hand, the discovery of HCO₃ as a receptor agonist is treated as serendipity, but on the other hand the first sections of the manuscript are presented as a concerted bioinformatics effort to find such a HCO₃-activated receptor. So, the reader may be left wondering how this came to pass: Was it simply good luck and careful observation? or good sleuthing based on an insightful hunch?*

Response: We apologize for the confusing framing of our manuscript. We selected GPR30 as a result of ‘*good sleuthing based on an insightful hunch*’, while we, fortunately, found bicarbonate-induced activation of GPR30 as a result of ‘*simply good luck and careful observation*’.

We previously reported the proton-sensing receptor G2A (⁶Murakami et al., 2004. doi: 10.1074/jbc.M406561200) and have been searching for another GPCR related to acid–base balance. In the search for acid–base balance-related GPCRs, we selected candidate GPCRs based on their specific expression in the stomach and pancreas, where acid or alkaline secretions immediately affect neighbouring cells in the microenvironment. From our 10 candidate GPCRs, 4 GPCRs were highly expressed in the brain. Only GPR30 was expressed in vascular and perivascular cells in the neurovascular unit (NVU), which primarily contributes to the pathophysiology of ischaemic stroke. Thus, we focused on GPR30.

Because GPR30 is a G-protein-coupled oestrogen receptor (GPER) that mediates the rapid non-genomic action of oestradiol, we first evaluated the non-genomic action of oestradiol through GPR30 (Fig. 1a, Supplementary Fig. 2a), where oestradiol did not activate GPR30. We subsequently came across GPR30 activation by the culture medium and identified that bicarbonate in the culture medium activated GPR30 based on the results in Fig. 1b-f.

For clarity, we have corrected the sentence: ‘This finding led us to hypothesize that **another acid–base balance-related GPCR** modulates signal transduction in response to dynamic changes **in the local acid–base balance**’ in lines 71–73. We have also modified the first paragraph of the Discussion section (lines 342–346).

We greatly appreciate the reviewer’s comments.

REFERENCES

1. Zha XM, Xiong ZG, Simon RP. pH and proton-sensitive receptors in brain ischemia. *J Cereb Blood Flow Metab* 42, 1349-1363 (2022).

2. Smith ML, von Hanwehr R, Siesjo BK. Changes in extra- and intracellular pH in the brain during and following ischemia in hyperglycemic and in moderately hypoglycemic rats. *J Cereb Blood Flow Metab* 6, 574-583 (1986).
3. Kawabata Y. Effect of tris-hydroxymethyl-aminomethane in arterial blood, brain and cerebrospinal fluid acidosis after total cerebral ischemia in dogs. *Kawasaki Medical Journal* 19, 25-35 (1993).
4. Choi I. SLC4A transporters. *Curr Top Membr* 70, 77-103 (2012).
5. Chesler M. Regulation and modulation of pH in the brain. *Physiol Rev* 83, 1183-1221 (2003).
6. Murakami N, Yokomizo T, Okuno T, Shimizu T. G2A is a proton-sensing G-protein-coupled receptor antagonized by lysophosphatidylcholine. *J Biol Chem* 279, 42484-42491 (2004).

Responses to Reviewer #2:

We greatly appreciate Reviewer #2 for taking the time to evaluate our manuscript. We provide point-by-point responses to the reviewer's comments. Unless otherwise specified, the figure, page, and line numbers refer to those in the revised manuscript. Modified text is shown in green. Text in red indicates modifications for multiple reviewers.

In this manuscript, authors have identified the first bicarbonate-sensing GPCR and suggest that pH resilience due to the bicarbonate buffering system modulates signal transduction via GPCRs. The local concentration of bicarbonate ions rapidly increases upon reperfusion, which activates GPR30-positive pericytes, leading to the inhibition of the rapid recovery of peripheral circulation. The manuscript is well written, easy to read, and within the scope of the submitted journal. However, additional experiments and information should be done to improve the quality of the manuscript. Some concerns are outlined below, hoping that it helps the authors strengthen the manuscript.

1. Local pH and ion homeostasis are dynamically changed, and could be influenced by the microenvironment. How would the authors to control other variances?

Response: Previous studies consistently demonstrated that ischaemia causes a significant change in the local acid–base balance in the brain: a decrease in pH (from 7.3 to 6.0–6.5), an increase in pCO₂ (from 40–50 to >100 mmHg), and a decrease in [HCO₃⁻] (from 20–26 to 11–12 mM extracellularly and 11–12 to 4–8 mM intracellularly). Importantly, reperfusion restores these deviations to the pre-ischemic state (¹Zha et al., 2022. doi: 10.1177/0271678X221089074; ²Smith et al., 1986. doi: 10.1038/jcbfm.1986.104; ³Chesler, 2003. doi: 10.1152/physrev.00010.2003). Ischemia and reperfusion also cause significant changes in ion homeostasis: sodium, potassium, ionized calcium, chloride, and glucose (⁷Martha et al., 2018. doi: 10.1016/j.jstrokecerebrovasdis.2018.05.045; ⁸Kumar et al., 2019. doi: 10.1088/1361-6579/ab47ee).

In this study, we did not artificially control the acid–base balance and ion homeostasis, and rapid changes in pH, electrolytes, and glucose are possibly involved in the pathophysiology of ischaemia-reperfusion injury, such as activation of mural cells. However, because none of these variances, other than bicarbonate ions, activated GPR30 *in vitro* (Fig. 1d, Supplementary Fig. 2e), it is most likely that the changes in the local bicarbonate concentration among other variances during ischaemia and reperfusion affected GPR30 activation and were involved in the ameliorated ischaemia-reperfusion injury in GPR30-deficient mice. We have discussed this issue in the Discussion section (lines 404–415).

2. *Why “10 of 353 GPCRs examined were selected based on their predominant expression in the stomach and pancreas”, but the object of this study is the brain?*

Response: We previously reported the proton-sensing receptor G2A (⁶Murakami et al., 2004. doi: 10.1074/jbc.M406561200) and have been searching for another GPCR related to acid–base balance. For clarity, we have corrected the sentence: ‘This finding led us to hypothesize that **another acid–base balance-related GPCR** modulates signal transduction in response to dynamic changes **in the local acid–base balance**’ in lines 71–73.

In the search for acid–base balance-related GPCRs, we chose 10 candidate GPCRs based on their specific expression in the stomach and pancreas, where acid or alkaline secretions immediately act on neighbouring cells in the microenvironment. Because ischaemia and reperfusion would directly affect the local acid–base balance due to blood supply, we employed the ischemia-reperfusion model in which rapid acid–base balance shifts were reported (⁹Ma et al., 2020. doi: 10.1016/j.biopha.2020.110686). Because 4 of the 10 candidate GPCRs, including GPR30, were highly expressed in the brain microvasculature, we focused on cerebral ischaemia-reperfusion and used the MCAO model, which primarily contributes to the pathophysiology of ischaemic stroke. We have revised the Introduction and Results sections and described this point in the text (lines 77–85, 97–99).

3. *Please clarify the ER-EGFP and mock-EGFP?*

Response: We apologize for inadvertently omitting the source of the ER-EGFP and mock-EGFP vectors. The ER-EGFP expression vector used in our research was a gift from the Takayanagi laboratory at Kyushu University (¹⁰Wu et al., 2006. doi: 10.1128/MCB.01534-05). The mock-EGFP vector was the pEGFP-N2 from Clontech. We have added the information in the Methods section (lines 485–486).

4. *The word size in Figures is too small to see. Please make all words bigger.*

Response: We sincerely apologize that the font size was too small to be seen in many figures. We have made it bigger in all the panels.

5. *In Figure 3, have the authors checked the GPR30 expression on SMCs?*

Response: Thank you very much for the constructive comment. To evaluate GPR30 expression on SMCs, we first quantified the expression of GPR30 in cerebral arteries using quantitative PCR. Although mRNA levels of GPR30 were slightly higher in cerebral arteries than in the whole brain

cortex, the mRNA expression profile showed contamination of other cell types, including endothelial cells, astrocytes, and oligodendrocytes, by the expression of their marker genes, *Cldn5* and *Tie2*, *Gfap*, and *Pdgfra*, respectively (data shown below).

We then performed in situ hybridization to determine if GPR30 is expressed in SMCs in addition to pericytes (Fig. 5f). The data demonstrated the expression of GPR30 in SMCs (*Acta2*⁺*Pdgfrb*⁺). Here, we also confirmed the expression of GPR30 in pericytes (*Acta2*⁺*Pdgfrb*⁺). We have described these points in the text (lines 226–229, 232, 251). We have also revised the subheading ‘Bicarbonate-GPR30 signal in **brain mural cells**’ in line 200 and the caption ‘GPR30 expressed in brain **mural cells** is activated by bicarbonate’ in Fig. 5.

Scale bar: 50 μm

6. The GPR30-deficient mice were utilized in this study. Will the pericyte conditional GPR30-deficient mice be more specific to explore the functions of GPR30?

Response: Because GPR30 is expressed in different cell types in different organs, we agree that pericyte-specific GPR30-deficient mice would be more suitable for analysing the cerebral ischemia-reperfusion model.

Per your suggestion, we have started establishing the *Gpr30*^{fllox/fllox} mouse. Because it takes a long time

to establish pericyte conditional GPR30-deficient mice, we have mentioned this issue as a limitation in the Discussion section (lines 434–436).

7. To show mature and functional neurons, was there synapse protein expression in the bioprinted neural tissue.

Response: Following your comment, we have evaluated the expression of neuronal, astrocytic, and synaptic markers using western blotting (Supplementary Fig. 7a, b) instead of analysing bioprinted neural tissues. The data showed a comparable expression of neuronal, astrocytic, and synaptic marker proteins. We concluded that there were no obvious defects in the neural maturation and function in GPR30-deficient mice. We have added these data in the Results section (lines 257–260) and Supplementary Fig. 7a, b.

We greatly appreciate the reviewer's comments.

REFERENCES

1. Zha XM, Xiong ZG, Simon RP. pH and proton-sensitive receptors in brain ischemia. *J Cereb Blood Flow Metab* 42, 1349-1363 (2022).
2. Smith ML, von Hanwehr R, Siesjo BK. Changes in extra- and intracellular pH in the brain during and following ischemia in hyperglycemic and in moderately hypoglycemic rats. *J Cereb Blood Flow Metab* 6, 574-583 (1986).
5. Chesler M. Regulation and modulation of pH in the brain. *Physiol Rev* 83, 1183-1221 (2003).
6. Murakami N, Yokomizo T, Okuno T, Shimizu T. G2A is a proton-sensing G-protein-coupled receptor antagonized by lysophosphatidylcholine. *J Biol Chem* 279, 42484-42491 (2004).
7. Martha SR, et al. Translational evaluation of acid/base and electrolyte alterations in rodent model of focal ischemia. *J Stroke Cerebrovasc Dis* 27, 2746-2754 (2018).
8. Kumar G, Kasiviswanathan U, Mukherjee S, Kumar Mahto S, Sharma N, Patnaik R. Changes in electrolyte concentrations alter the impedance during ischemia-reperfusion injury in rat brain. *Physiol Meas* 40, 105004 (2019).
9. Ma R, et al. Animal models of cerebral ischemia: A review. *Biomed Pharmacother* 131, 110686 (2020).
10. Wu Y, Kawate H, Ohnaka K, Nawata H, Takayanagi R. Nuclear compartmentalization of N-CoR and its interactions with steroid receptors. *Mol Cell Biol* 26, 6633-6655 (2006).

Responses to Reviewer #3:

We greatly appreciate Reviewer #3 for taking the time to evaluate our manuscript. We are especially grateful for the comment *'The data reported may prove to be highly significant and suggest a novel druggable target for the treatment of ischaemic stroke'*. We provide point-by-point responses to the reviewer's comments. Unless otherwise specified, the figure, page, and line numbers refer to those in the revised manuscript. Modified text is shown in cyan blue. Text in red indicates modifications for multiple reviewers.

The manuscript by Jo-Watanabe and colleagues describes the results of the experimental studies collectively suggesting that bicarbonate signalling via GPR30 expressed by pericytes contributes to ischaemia-reperfusion brain injury. The authors propose that ischemia is associated with an increase in extracellular bicarbonate, which acts at GPR30 expressed by pericytes, leading to Gq mediated elevation of intracellular calcium, pericyte constriction, ultimately preventing successful reperfusion of the affected brain region. The study appears to be expertly performed, nicely illustrated and the text of the manuscript is well written. The data reported may prove to be highly significant and suggest a novel druggable target for the treatment of ischaemic stroke. I only have two comments to make which the authors may consider in their revision:

- 1. Fig 1g shows that the responses (IP accumulation) mediated by GPR30 expressed in HEK293 cells are pretty much saturated at concentrations above 20 mM. Fig 2a shows similar responses (Ca²⁺ mobilization) to 22 and 33 mM HCO₃⁻ in mouse myoblasts. Fig 3i shows robust Ca²⁺ responses in brain pericytes triggered by 11 and 22 mM bicarbonate. If these observations are correct, then GPR30 expressed by pericytes would be constitutively active at physiological concentrations of extracellular HCO₃⁻ in the brain (22-26 mM). Fig 5b,c shows a very modest increase (by ~4 mM) in bicarbonate measured in the blood collected from the facial vein. The authors need to discuss how they envisage the operation of the proposed mechanism as the receptor is clearly strongly activated by bicarbonate at its physiological concentration and ischaemia/reperfusion-induced increases in HCO₃⁻ are too small to trigger functionally significant further increases in pericyte Ca²⁺.*

Response: We acknowledge the validity of your concern. We recognize that bicarbonate-induced activation of GPR30 *in vivo* requires shifting local bicarbonate levels over the dynamic range, as demonstrated in Fig. 2h and i of the revised manuscript. The change in the systemic, i.e., serum, bicarbonate levels does not precisely mirror the change in local bicarbonate levels in the brain, although the bicarbonate concentration in the serum sometimes reflects that in the brain. The systemic

attenuation of the local dynamic shift in the bicarbonate concentration is attributed to the bicarbonate-buffering system *in vivo*.

Previous studies consistently demonstrated that ischaemia causes a significant change in the local acid–base balance in the brain: a decrease in pH (from 7.3 to 6.0–6.5), an increase in pCO₂ (from 40–50 to >100 mmHg), and a decrease in [HCO₃⁻] (from 20–26 to 11–12 mM extracellularly and 11–12 to 4–8 mM intracellularly). Importantly, reperfusion restored these deviations to the pre-ischemic state (¹Zha et al., 2022. doi: 10.1177/0271678X221089074; ²Smith et al., 1986. doi: 10.1038/jcbfm.1986.104; ³Kawabata, 1993. doi: 10.11482/KMJ19(1)25-35.1993.pdf).

Figure A1. Change in cerebrospinal fluid (CSF) pH, PCO₂, and HCO₃⁻.

Each point indicates the mean ± SE for six dogs.
#: Significantly different from pre-ischemic value.
(p<0.05)

Figure A2. Change in arterial pH, PCO₂, and HCO₃⁻.

Each point indicates the mean ± SE for six dogs.
#: Significantly different from pre-ischemic value.
(p<0.05)

Adapted from Kawabata Y, 'Effect of Tris-hydroxymethyl-aminomethane on Arterial Blood, Brain and Cerebrospinal Fluid Acidosis after Total Cerebral Ischemia in Dogs', *Kawasaki Medical Journal*, 1993.

Contrastingly, the systemic pH, pCO₂, and [HCO₃⁻] do not shift as much as those in the local microenvironment. Previous studies which parallely evaluated CSF/tissue and blood bicarbonate concentrations demonstrated that the CSF/tissue bicarbonate concentrations decreased to 11–12 mM, equivalent to the EC₅₀ in our *in vitro* experiment, during ischaemia and recovered to the preischemic level by reperfusion, whereas ischaemia caused a slight decrease (~3 mM) in the blood bicarbonate concentrations (²Smith et al., 1986. Doi: 10.1038/jcbfm.1986.104; ³Kawabata, 1993. Doi: 10.11482/KMJ19(1)25-35.1993.pdf, **Figure A1 and A2**). Therefore, in our experiment, it is most likely that the extracellular bicarbonate concentration decreased to the EC₅₀ of bicarbonate-induced GPR30 activation (11–12 mM) during ischaemia and recovered enough for full activation (20–26 mM) after reperfusion, while the serum bicarbonate concentration increased slightly but significantly from 22 to 26 mM. **Consequently, changes in the local bicarbonate concentration during ischaemia and reperfusion likely affected GPR30 activation *in vivo*.** It should be noted, however, that we cannot exclude the possibilities that the *in vivo* dynamic range of GPR30 activation by bicarbonate is different from that *in vitro* and that constitutive activation of GPR30 is involved in the pathophysiology

of ischaemia-reperfusion injury.

The above-mentioned studies used rat or canine ischaemia-reperfusion models and calculated $[\text{HCO}_3^-]$ from the pH and pCO_2 values that were measured using electrodes inserted in the brain. Because direct measurement of extracellular bicarbonate concentrations in mice would be informative, we attempted to directly quantify local changes in the bicarbonate concentration using a microdialysis system in the mouse MCAO model. *In vitro* recovery tests demonstrated that the bicarbonate concentration of the recovered fluid mirrored the local concentration outside the dialysis membrane (**Figure B1**). However, the recovered fluid from the MCAO mice consistently contained the same concentration of bicarbonate as the dialysate over a range of bicarbonate concentrations (**Figure B2**), indicating that our microdialysis system failed to reflect the local bicarbonate concentration *in vivo*.

Figure B1. *In vitro* recovery test.

A microdialysis probe was sequentially inserted into the tubes containing different sodium bicarbonate concentrations. Bicarbonate concentrations of dialysate, sodium bicarbonate solutions in the tubes (outside), and recovered fluid were measured.

Figure B2. *In vivo* microdialysis.

A microdialysis probe was inserted into the left striatum. Dialysates containing different sodium bicarbonate concentrations were continuously infused, and recovered fluid was collected. Bicarbonate concentrations of dialysate and recovered fluid were measured.

In the revised manuscript, we have explicitly acknowledged that we did not measure the local bicarbonate concentration as a limitation of this study in the Discussion section. We have also discussed whether the changes in $[\text{HCO}_3^-]$ in the brain could be in the appropriate range to cause GPR30 activation (line 389–403, 416–420).

2. *I suggest that authors discuss the potential mechanisms leading to the increased concentration of extracellular bicarbonate during ischaemia/reperfusion.*

Response: Previous studies consistently demonstrated that ischaemia causes a significant change in the local acid–base balance in the brain: a decrease in pH, an increase in pCO_2 , and a decrease in $[\text{HCO}_3^-]$, which are restored to the pre-ischemic state by reperfusion.

The restricted blood flow limits the supply of oxygen to the tissues. Accordingly, the cellular metabolic state shifts from aerobic respiration to anaerobic glycolysis, producing lactate that causes intracellular acidosis. Consequently, the tissue pH decreases to 6.0–6.5 within minutes of ischemia (²Smith et al., 1986. doi: 10.1038/jcbfm.1986.104; ¹¹Nemoto et al., 1981. doi: 10.1161/01.str.12.1.77; ¹²Mabe et al., 1983. doi: 10.1038/jcbfm.1983.13; ¹³Hoffman et al., 1997. doi: 10.1093/bja/78.2.169). The following mechanisms to counteract intracellular acidosis cause interstitial acidosis. One is the subsequent lactate export via monocarboxylate transporters, resulting in extracellular metabolic acidosis. Other mechanisms include activation of the Na⁺/H⁺ exchanger, NHE, and import of bicarbonate ions to counteract intracellular acidification, via SLC4A family transporters: sodium bicarbonate cotransporter NBC, Na⁺/Cl⁻/2HCO₃⁻ cotransporter/exchanger NDCBE, and Cl⁻/HCO₃⁻ exchanger AE (⁴Choi, 2012. doi: 10.1016/B978-0-12-394316-3.00003-X). **All these mechanisms contribute to the reduction of extracellular bicarbonate levels** (⁵Chesler, 2003. doi: 10.1152/physrev.00010.2003).

According to your comments, we have described potential mechanisms leading to the increased concentration of extracellular bicarbonate during ischaemia/reperfusion in the Discussion section (lines 404–410).

We greatly appreciate the reviewer's comments.

REFERENCES

1. Zha XM, Xiong ZG, Simon RP. pH and proton-sensitive receptors in brain ischemia. *J Cereb Blood Flow Metab* 42, 1349-1363 (2022).
2. Smith ML, von Hanwehr R, Siesjo BK. Changes in extra- and intracellular pH in the brain during and following ischemia in hyperglycemic and in moderately hypoglycemic rats. *J Cereb Blood Flow Metab* 6, 574-583 (1986).
3. Kawabata Y. Effect of Tris-hydroxymethyl-aminomethane in arterial blood, brain and cerebrospinal fluid acidosis after total cerebral ischemia in dogs. *Kawasaki Medical Journal* 19, 25-35 (1993).
4. Choi I. SLC4A transporters. *Curr Top Membr* 70, 77-103 (2012).
5. Chesler M. Regulation and modulation of pH in the brain. *Physiol Rev* 83, 1183-1221 (2003).
11. Nemoto EM, Frinak S. Brain tissue pH after global brain ischemia and barbiturate loading in rats. *Stroke* 12, 77-82 (1981).
12. Mabe H, Blomqvist P, Siesjo BK. Intracellular pH in the brain following transient ischemia. *J Cereb Blood Flow Metab* 3, 109-114 (1983).
13. Hoffman WE, Charbel FT, Edelman G, Ausman JI. Brain tissue oxygenation in patients with cerebral occlusive disease and arteriovenous malformations. *Br J Anaesth* 78, 169-171 (1997).

Responses to Reviewer #4:

We greatly appreciate **Reviewer #4** for taking the time to evaluate our manuscript. We are especially grateful for the comments '*This is an excellent study which identifies the bicarbonate sensing role of GPR30 in pericytes as a key modulator of blood flow in ischemia reperfusion*'. We provide point-by-point responses to the reviewer's comments. Unless otherwise specified, the figure, page, and line numbers refer to those in the revised manuscript. Modified text is shown in purple. Text in red indicates modifications for multiple reviewers.

This is an excellent study which identifies the bicarbonate sensing role of GPR30 in pericytes as a key modulator of blood flow in ischemia reperfusion. The systematic testing which establishes the bicarbonate response of the receptor while ruling out other potential contributors is quite beautifully done. I have several comments that I hope the reviewers will address to strengthen the manuscript.

1. As the authors acknowledge, GPR30 is also highly expressed in smooth muscle cells, which is evident in the RNAseq database they utilize and likely also shown in their images of GPR30 localization. It is important to delineate responses of pericytes vs. SMCs to bicarbonate levels with higher resolution to determine relative contributions to bicarbonate fluctuations in vivo. This could be achieved by imaging calcium in these cells either isolated from the brain and loaded with calcium dyes, or in cells isolated from mice expressing GCaMP under the NG2 or pdgfrb promoters. The smooth muscle may play an important role here that is overlooked, as changes in bicarbonate during I-R injury are likely to cover large areas which encapsulate both pericytes on capillaries and SMCs on arterioles.

Response: Thank you very much for the constructive comment.

To evaluate GPR30 expression on SMCs, we first quantified the expression of GPR30 in cerebral arteries using quantitative PCR. Although mRNA levels of GPR30 were slightly higher in cerebral arteries than in the whole brain cortex, the mRNA expression profile showed contamination of other cell types, including endothelial cells, astrocytes, and oligodendrocytes, by the expression of their marker genes, *Cldn5* and *Tie2*, *Gfap*, and *Pdgfra*, respectively (data shown below).

Because the quantitative PCR data did not allow us to draw a firm conclusion about GPR30 expression on SMCs, we next performed in situ hybridization to determine if GPR30 is expressed in SMCs in addition to pericytes (Fig. 5f). The data demonstrated the expression of GPR30 in SMCs (*Acta2*⁺*Pdgfrb*⁺). Here, we also confirmed the expression of GPR30 in pericytes (*Acta2*⁺*Pdgfrb*⁺). We have described these points in the text (lines 226–229, 232, 251). We have also revised the subheading ‘Bicarbonate-GPR30 signal in **brain mural cells**’ in line 200 and the caption ‘GPR30 expressed in brain **mural cells** is activated by bicarbonate’ in Fig. 5.

Scale bar: 50 μm

We then performed calcium imaging in freshly isolated SMCs and pericytes independently isolated from the conditional GCaMP knock-in mice expressing GCaMP under the *Gpr30* promoter (Supplementary Fig. 6a, lines 672–679 in the Methods section). We found a GPR30-dependent bicarbonate-induced increase in calcium levels in SMCs and pericytes (Fig. 5m-r). This was consistent with the increased blood flow recovery in GPR30-deficient mice as shown using the laser Doppler flowmetry and MRA, which detected blood flow in capillaries and arteries, respectively. Collectively, these results demonstrated that GPR30 in both SMCs and pericytes senses bicarbonate to activate intracellular signalling cascades.

We have described these results in the text (lines 243–250, 251, 338–339) and in Fig. 5f and m-r. We have also discussed the future directions derived from these results in the Discussion section (lines 421–432). We have also added ‘*Fresh isolation of mouse brain vascular smooth muscle cells and pericytes*’ subsection in the Methods section (lines 737–745).

2. *The use of primary culture to demonstrate the bicarbonate signal in brain pericytes is not ideal, as phenotypic drift may affect responses. Use of an acute isolation protocol e.g. PMID: 35349300 would increase confidence that activation of native GPR30 in pericytes by bicarbonate ions is a physiologically relevant event.*

Response: Thank you very much for your valuable comment. We agree that using a primary culture is not ideal, as phenotypic drift may affect the responses. In response to your comments 1 and 2, we performed an acute isolation protocol (¹⁴Sancho et al., 2022. Doi: 10.1126/scisignal.ab15405) followed by Ca imaging using pericytes and SMCs. We found a GPR30-dependent bicarbonate-induced increase in calcium levels in freshly isolated SMCs and pericytes (Fig. 5m-r). These results demonstrated that bicarbonate-GPR30 signalling is shared between SMCs and pericytes. As you suggested, bicarbonate-induced GPR30 activation in freshly isolated mural cells has greatly increased the certainty that activation of native GPR30 in pericytes by bicarbonate ions is a physiologically relevant event. We have described these results in lines 243–250 and added Fig. 5m-r. We have also added ‘*Fresh isolation of mouse brain vascular smooth muscle cells and pericytes*’ subsection in the Methods section (lines 737–745).

3. *Minor comment: At times figure labeling is unclear, and a careful check that all terms are defined in the figure legends is important for readability/interpretability. For example, Ex Fig 7d-f have an 'NT' condition which is not defined which makes interpretation difficult.*

Response: We sincerely apologize that the figure labelling was unclear and that some terms were not defined in the figure legends. We have carefully checked the figures and legends, made the font size larger, and added the definitions of abbreviations, including that of 'NT' in the Supplementary Fig. 7f-h in the revised manuscript.

4. *Minor comment: A number of figures also have a confusing star denoting significant differences between groups in the legend, but then testing on the data show no difference. For example Ex Fig 7b has a star between group labels, but then both tests on the data show 'ns' for not significant. This issue is present in several figures and creates confusion. Clarification in the figure legends or some other approach is needed to avoid confusing the reader.*

Response: We sincerely apologize that many figures included confusing stars, especially Extended Data Fig. 7b in the previous manuscript (Supplementary Fig. 7d in the revised manuscript). For clarity, we have omitted the star between *Gpr30^{+/Venus}* and *Gpr30^{Venus/Venus}* mice in Supplementary Fig. 7d and only mentioned as 'although with slightly higher endothelial coverage in *Gpr30^{-Venus}* mice than in *Gpr30^{+/Venus}* mice' in lines 265-267.

We greatly appreciate the reviewer's comments.

REFERENCES

14. Sancho M, et al. Adenosine signaling activates ATP-sensitive K(+) channels in endothelial cells and pericytes in CNS capillaries. *Sci Signal* 15, eab15405 (2022).

REFERENCES

1. Zha XM, Xiong ZG, Simon RP. pH and proton-sensitive receptors in brain ischemia. *J Cereb Blood Flow Metab* **42**, 1349-1363 (2022).
2. Smith ML, von Hanwehr R, Siesjo BK. Changes in extra- and intracellular pH in the brain during and following ischemia in hyperglycemic and in moderately hypoglycemic rats. *J Cereb Blood Flow Metab* **6**, 574-583 (1986).
3. Kawabata Y. Effect of Tris-hydroxymethyl-aminomethane on Arterial Blood, Brain and Cerebrospinal Fluid Acidosis after Total Cerebral Ischemia in Dogs. *Kawasaki Medical Journal* **19**, 25-35 (1993).
4. Choi I. SLC4A transporters. *Curr Top Membr* **70**, 77-103 (2012).
5. Chesler M. Regulation and modulation of pH in the brain. *Physiol Rev* **83**, 1183-1221 (2003).
6. Murakami N, Yokomizo T, Okuno T, Shimizu T. G2A is a proton-sensing G-protein-coupled receptor antagonized by lysophosphatidylcholine. *J Biol Chem* **279**, 42484-42491 (2004).
7. Martha SR, *et al.* Translational evaluation of acid/base and electrolyte alterations in rodent model of focal ischemia. *J Stroke Cerebrovasc Dis* **27**, 2746-2754 (2018).
8. Kumar G, Kasiviswanathan U, Mukherjee S, Kumar Mahto S, Sharma N, Patnaik R. Changes in electrolyte concentrations alter the impedance during ischemia-reperfusion injury in rat brain. *Physiol Meas* **40**, 105004 (2019).
9. Ma R, *et al.* Animal models of cerebral ischemia: A review. *Biomed Pharmacother* **131**, 110686 (2020).
10. Wu Y, Kawate H, Ohnaka K, Nawata H, Takayanagi R. Nuclear compartmentalization of N-CoR and its interactions with steroid receptors. *Mol Cell Biol* **26**, 6633-6655 (2006).
11. Nemoto EM, Frinak S. Brain tissue pH after global brain ischemia and barbiturate loading in rats. *Stroke* **12**, 77-82 (1981).

12. Mabe H, Blomqvist P, Siesjo BK. Intracellular pH in the brain following transient ischemia. *J Cereb Blood Flow Metab* **3**, 109-114 (1983).
13. Hoffman WE, Charbel FT, Edelman G, Ausman JI. Brain tissue oxygenation in patients with cerebral occlusive disease and arteriovenous malformations. *Br J Anaesth* **78**, 169-171 (1997).
14. Sancho M, *et al.* Adenosine signaling activates ATP-sensitive K(+) channels in endothelial cells and pericytes in CNS capillaries. *Sci Signal* **15**, eabl5405 (2022).

REVIEWERS' COMMENTS

Reviewer #1 (Remarks to the Author):

This revised manuscript has been adequately responsive to previous criticisms. It is worth noting that the role of HCO₃ per se as the in vivo agonist for GPR30 effects on ischemia-reperfusion injury remains to be directly established. However, the authors have acknowledged this limitation and provided a plausible and evidence-based description for how decreases in HCO₃ during ischemia and subsequent increases in HCO₃ during reperfusion could traverse the activation range for GPR30 to account for receptor contributions to injury. The paper continues to provide a strong case for the novel idea that GPR30 is an HCO₃-activated receptor. This reviewer has no more comments.

Reviewer #2 (Remarks to the Author):

You did great revision.

Reviewer #3 (Remarks to the Author):

The manuscript by Jo-Watanabe and colleagues describes the results of the experimental studies suggesting that GPR30 expressed by mural cells (the authors focus on brain pericytes in particular) is sensitive to bicarbonate and contributes to ischaemia-reperfusion brain injury.

The authors now provided detailed response to my comments on their first submission and, unfortunately, I now found the case for the key role played by bicarbonate-induced activation of GPR30 in the pathogenesis of the ischaemic stroke less compelling. In their rebuttal letter the authors reproduce the data from a different publication, actually showing that during brain ischaemia there is a reduction in extracellular HCO₃⁻ (from 20-26 mM to ~11-12 mM), which slowly recovers to the normal level during reperfusion. Therefore, there is no evidence to suggest that extracellular bicarbonate concentration is increasing during stroke. The data reported in the paper show that GPR30 expressed by pericytes would be constitutively active at physiological concentrations of extracellular HCO₃⁻ in the brain (22-26 mM). I am struggling to envisage the operation of the proposed mechanism as the receptor is clearly strongly activated by bicarbonate at its physiological concentration and it would be expected to be less active during stroke. Considering that the dataset obtained in GPR30 deficient mice is very convincing, my suggestion would be to revise the text thoroughly, clearly saying that the study identified a bicarbonate-sensitive GPCR (a significant advance on its own) and that this receptor facilitates the ischaemia/reperfusion brain injury. In my opinion, the claim that GPR30 activation by bicarbonate release during ischaemia/reperfusion is not tenable based on the data presented.

Other comments:

1. Page 2: The authors say "Mechanistically, the local concentration of bicarbonate ions rapidly increases upon reperfusion, which activates GPR30-positive pericytes, inhibiting the rapid recovery of peripheral circulation". This is an example of a key conclusion included in the text of the abstract which is not supported by the reported data. There is no data presented to suggest that local concentration of bicarbonate ions rapidly increases upon reperfusion above the baseline and then activates GPR30-positive pericytes to reduce perfusion. Also the term "peripheral circulation" is often used to describe the circulation of blood in the peripheral tissues, which is obviously separate from cerebral circulation.
2. Page 3: "...and provide perspectives on physiological resilience in buffering systems that supply ligands for receptors". I am not sure that I understand this conclusion.
3. Page 3: "Alternatively, recent studies have demonstrated that bicarbonate and CO₂ directly regulate various cellular responses independently of pH_{11, 12}". I think the authors should acknowledge a major contribution made by Prof Dale to our understanding of direct CO₂ sensing in the brain (PMID: 24220509; PMID: 35102601).

Reviewer #4 (Remarks to the Author):

Thank you for your constructive and detailed responses to my comments. These have indeed strengthened the paper. I have several minor residual concerns which should be readily addressable.

1. The data added to Figure 5 are illuminating and a solid addition to the paper. I recommend adding labelling for m-r to clearly indicate whether the measurement is being made in SMCs or pericytes for the ease of the reader. The y axis in 5q could also be rescaled to make the response clearer.
2. The discussion of the results in f5 (lines 421-432) would benefit from additional attention to the wording used. As there are at least two recognized forms of pericytes (contractile/ensheathing and thin-strand), explicitly stating which form of these cells is thought to contract in these instances would be beneficial for clarity.
3. For the data in Supp f7d, the clarifications indicate that there is no significant difference between the data presented. Accordingly, the authors cannot state that endothelial coverage is even slightly higher in lines 265-267 referring to these data. If there is no statistical significance between the groups, no conclusions about any differences for be drawn.

Responses to the reviewers' comments on 'Bicarbonate signalling via GPCR regulates ischaemia-reperfusion injury' by Jo-Watanabe et al. (NCOMMS-23-03129-A).

Responses to Reviewer #1

We greatly appreciate Reviewer #1 for taking the time to evaluate our manuscript.

This revised manuscript has been adequately responsive to previous criticisms. It is worth noting that the role of HCO₃ per se as the in vivo agonist for GPR30 effects on ischemia-reperfusion injury remains to be directly established. However, the authors have acknowledged this limitation and provided a plausible and evidence-based description for how decreases in HCO₃ during ischemia and subsequent increases in HCO₃ during reperfusion could traverse the activation range for GPR30 to account for receptor contributions to injury. The paper continues to provide a strong case for the novel idea that GPR30 is an HCO₃-activated receptor. This reviewer has no more comments.

Response: We greatly appreciate the reviewer's comments and the reviewer for commending our work.

Responses to Reviewer #2:

We greatly appreciate Reviewer #2 for taking the time to evaluate our manuscript.

You did great revision.

Response: We are happy that our revisions are satisfactory and thank the reviewer for commending our work.

Responses to Reviewer #3:

We greatly appreciate Reviewer #3 for taking the time to evaluate our manuscript. We have provided point-by-point responses to the reviewer's comments below. Unless otherwise specified, the figure, page, and line numbers refer to those in the revised manuscript. Modified text is highlighted in cyan.

The manuscript by Jo-Watanabe and colleagues describes the results of the experimental studies suggesting that GPR30 expressed by mural cells (the authors focus on brain pericytes in particular) is sensitive to bicarbonate and contributes to ischaemia-reperfusion brain injury.

The authors now provided detailed response to my comments on their first submission and, unfortunately, I now found the case for the key role played by bicarbonate-induced activation of GPR30 in the pathogenesis of the ischaemic stroke less compelling. In their rebuttal letter the authors reproduce the data from a different publication, actually showing that during brain ischaemia there is a reduction in extracellular HCO₃⁻ (from 20-26 mM to ~11-12 mM), which slowly recovers to the normal level during reperfusion. Therefore, there is no evidence to suggest that extracellular bicarbonate concentration is increasing during stroke. The data reported in the paper show that GPR30 expressed by pericytes would be constitutively active at physiological concentrations of extracellular HCO₃⁻ in the brain (22-26 mM). I am struggling to envisage the operation of the proposed mechanism as the receptor is clearly strongly activated by bicarbonate at its physiological concentration and it would be expected to be less active during stroke. Considering that the dataset obtained in GPR30 deficient mice is very convincing, my suggestion would be to revise the text thoroughly, clearly saying that the study identified a bicarbonate-sensitive GPCR (a significant advance on its own) and that this receptor facilitates the ischaemia/reperfusion brain injury. In my opinion, the claim that GPR30 activation by bicarbonate release during ischaemia/reperfusion is not tenable based on the data presented.

Response: We appreciate the reviewer's comments and suggestions. We agree with the concern that we have not got convincing data for our proposed mechanism, i.e., the lack of demonstration of deactivation of GPR30 due to a decrease in local bicarbonate concentrations during ischaemia and reactivation due to an increase in local bicarbonate concentrations during transitions from ischaemia to reperfusion. Also, we cannot exclude the possibility that GPR30 might be constitutively active throughout ischaemia and reperfusion, without demonstration of dynamic shifts in local bicarbonate levels in our ischaemia-reperfusion model.

Therefore, in the revised manuscript, following the reviewer's suggestion, we toned down our claim regarding the mechanism that rapid shifts of bicarbonate concentrations during ischaemia-

reperfusion activate GPR30 and lead to ischaemia–reperfusion injury, referring to the lack of convincing data. **We have thoroughly revised the manuscript, removed unsupported claims particularly from the abstract, and emphasised the speculative nature of this proposed mechanism in the discussion section.** We have also revised the manuscript to clarify the major findings of our study: identification of a bicarbonate-sensing GPCR that facilitates cerebral ischaemia–reperfusion injury. The revisions related to this point are highlighted in cyan.

Other comments:

1. *Page 2: The authors say "Mechanistically, the local concentration of bicarbonate ions rapidly increases upon reperfusion, which activates GPR30-positive pericytes, inhibiting the rapid recovery of peripheral circulation". This is an example of a key conclusion included in the text of the abstract which is not supported by the reported data. There is no data presented to suggest that local concentration of bicarbonate ions rapidly increases upon reperfusion above the baseline and then activates GPR30-positive pericytes to reduce perfusion. Also the term "peripheral circulation" is often used to describe the circulation of blood in the peripheral tissues, which is obviously separate from cerebral circulation.*

Response: We agree with the concern that our claim *"Mechanistically, the local concentration of bicarbonate ions rapidly increases upon reperfusion, which activates GPR30-positive pericytes, inhibiting the rapid recovery of peripheral circulation"* is not supported by the presented data. Therefore, we have deleted this sentence from the manuscript. Addressing to the reviewer's comment on the usage of "peripheral circulation", we have now used "microcirculation" when we refer to blood flow in the brain microvasculature.

2. *Page 3: "...and provide perspectives on physiological resilience in buffering systems that supply ligands for receptors". I am not sure that I understand this conclusion.*

Response: We apologise for the confusion. This conclusion is based on the background described in the introduction section that several GPCRs have been identified as proton-sensing GPCRs, whereas GPR30 is the newly identified bicarbonate-sensing GPCR. Thus, these acid–base balance-related GPCRs are activated to concomitantly modulate cellular responses depending on proton and bicarbonate concentrations, i.e., the local acid–base balance. We have revised the last sentences of the abstract and discussion.

3. *Page 3: "Alternatively, recent studies have demonstrated that bicarbonate and CO₂ directly regulate various cellular responses independently of pH^{11, 12}". I think the authors should acknowledge a major contribution made by Prof Dale to our understanding of direct CO₂ sensing*

in the brain (PMID: 24220509; PMID: 35102601).

Response: As per the reviewer's suggestion, we have referenced the major contribution (PMID: 24220509) made by Prof. Dal¹, which is original research focusing on the pH-independent regulation of connexin 26 by CO₂.

REFERENCES

1. Meigh L, Greenhalgh SA, Rodgers TL, Cann MJ, Roper DI, Dale N. CO₂ directly modulates connexin 26 by formation of carbamate bridges between subunits. *Elife* **2**, e01213 (2013).

Responses to Reviewer #4:

We greatly appreciate Reviewer #4 for taking the time to evaluate our manuscript. We have provided point-by-point responses to the reviewer's comments below. Unless otherwise specified, the figure, page, and line numbers refer to those in the revised manuscript. Modified text is highlighted in green.

Thank you for your constructive and detailed responses to my comments. These have indeed strengthened the paper. I have several minor residual concerns which should be readily addressable.

1. *The data added to Figure 5 are illuminating and a solid addition to the paper. I recommend adding labelling for m-r to clearly indicate whether the measurement is being made in SMCs or pericytes for the ease of the reader. The y axis in 5q could also be rescaled to make the response clearer.*

Response: We thank the reviewer for the constructive suggestions. We have added labels for m-r (Fig. 6d-i in the revised manuscript) to indicate the cell type analysed. We have not rescaled the y axis in Fig. 5q (Fig. 6h in the revised manuscript), because it is aligned with that in Fig. 5p (Fig. 6g in the revised manuscript) to facilitate a comparison between $Gpr30^{+/iCre}$ and $Gpr30^{iCre/iCre}$ pericytes.

2. *The discussion of the results in f5 (lines 421-432) would benefit from additional attention to the wording used. As there are at least two recognized forms of pericytes (contractile/ensheathing and thin-strand), explicitly stating which form of these cells is thought to contract in these instances would be beneficial for clarity.*

Response: We thank the reviewer for this comment. As per the suggestion, we have described the morphological and functional characteristics of the two types of pericytes in the discussion section (lines 441–443). The *in situ* hybridisation assay (Fig. 5f) demonstrated that $Gpr30$ was expressed in both $Acta2^{+}Pdgfrb^{+}$ (SMCs and contractile/ensheathing pericytes) and $Acta2^{-}Pdgfrb^{+}$ (thin-strand pericytes) mural cells. Although GPR30 in contractile/ensheathing pericytes is likely to contribute to pericyte contraction leading to a reduction in blood flow, we did not distinguish the two forms of pericytes in the present study. Thus, bicarbonate-GPR30 signalling in each type of pericyte will be investigated in future studies. We have added this information to the discussion section (line 443–445).

3. *For the data in Supp f7d, the clarifications indicate that there is no significant difference between the data presented. Accordingly, the authors cannot state that endothelial coverage is even slightly higher in lines 265-267 referring to these data. If there is no statistical significance between the groups, no conclusions about any differences for be drawn.*

Response: We apologise that we mentioned a slightly higher endothelial coverage in lines 267–269, referring to Supplementary Fig. 7d, although no significant difference was indicated between the data presented. Actually, there was a significant difference between *Gpr30*^{+Venus} and *Gpr30*^{-Venus} mice, but the difference was no longer significant when analysed in the septal area and cortex subgroups. In the revised manuscript, we have indicated the statistical significance between *Gpr30*^{+Venus} and *Gpr30*^{-Venus} mice and stated this in the figure legend.

REFERENCES

1. Meigh L, Greenhalgh SA, Rodgers TL, Cann MJ, Roper DI, Dale N. CO₂ directly modulates connexin 26 by formation of carbamate bridges between subunits. *Elife* **2**, e01213 (2013).